# Structure, interaction and nervous connectivity of beta cell primary cilia

Andreas Müller [1,2,3,13] ✉, Nikolai Klena [4,13], Song Pang [5,10], Leticia Elizabeth Galicia Garcia[1,2,3,6], Oleksandra Topcheva [1,2,3], Solange Aurrecoechea Duran[1,2,3], Davud Sulaymankhil[1,2,3,11], Monika Seliskar[1,2,3], Hassan Mziaut[1,2,3], Eyke Schöniger[1,2,3], Daniela Friedland[1,2,3], Nicole Kipke[1,2,3], Susanne Kretschmar[7], Carla Münster[1,2,3], Jürgen Weitz[8], Marius Distler[8], Thomas Kurth [7], Deborah Schmidt[9], Harald F. Hess [5], C. Shan Xu [5,12], Gaia Pigino [4,13] ✉ & Michele Solimena [1,2,3,6,13] ✉

Primary cilia are sensory organelles present in many cell types, partaking in various signaling processes. Primary cilia of pancreatic beta cells play pivotal roles in paracrine signaling and their dysfunction is linked to diabetes. Yet, the structural basis for their functions is unclear. We present three-dimensional reconstructions of beta cell primary cilia by electron and expansion microscopy. These cilia are spatially confined within deep ciliary pockets or narrow spaces between cells, lack motility components and display an unstructured axoneme organization. Furthermore, we observe a plethora of beta cell cilia-cilia and cilia-cell interactions with other islet and non-islet cells. Most remarkably, we have identified and characterized axo-ciliary synapses between beta cell cilia and the cholinergic islet innervation. These findings highlight the beta cell cilia's role in islet connectivity, pointing at their function in integrating islet intrinsic and extrinsic signals and contribute to understanding their significance in health and diabetes.

Pancreatic islet beta cells control blood glucose levels by secreting the hormone insulin. Insulin secretion is a concerted process regulated by autocrine signaling cascades, paracrine interactions with other beta and non-beta pancreatic islet cells as well as factors of systemic and neuronal origin[1-3]. The specific architecture of the islets plays a significant role in the orchestration of these pathways, as these mini-organs are highly vascularized[4], with their endocrine cells forming rosettes around the capillaries[5]. Furthermore, islets make multiple connections with the nervous system[3]. Most beta cells possess a single primary cilium projecting from the cell's a-vascular pole[6], which has been detected using various microscopy methods[7-13]. Cilia contribute to paracrine signal transduction within the islets of Langerhans[14-17]. It has been shown that beta cell cilia play a pivotal role in glucose homeostasis and their loss or dysfunction impairs insulin secretion and

[1]Molecular Diabetology, University Hospital and Faculty of Medicine Carl Gustav Carus, TU Dresden, Dresden, Germany. [2]Paul Langerhans Institute Dresden (PLID) of Helmholtz Munich, University Hospital Carl Gustav Carus and Faculty of Medicine, TU Dresden, Dresden, Germany. [3]German Center for Diabetes Research (DZD e.V.), Neuherberg, Germany. [4]Human Technopole (HT), Milan, Italy. [5]Janelia Research Campus, Howard Hughes Medical Institute, Ashburn, VA, USA. [6]DFG Cluster of Excellence "Physics of Life", TU Dresden, Dresden, Germany. [7]Center for Molecular and Cellular Bioengineering (CMCB), Technology Platform, Core Facility Electron Microscopy and Histology, TU Dresden, Dresden, Germany. [8]Department of Visceral, Thoracic and Vascular Surgery, University Hospital Carl Gustav Carus, Medical Faculty, TU Dresden, Dresden, Germany. [9]HELMHOLTZ IMAGING, Max Delbrück Center for Molecular Medicine (MDC) in the Helmholtz Association, Berlin, Germany. [10]Present address: Yale School of Medicine, New Haven, CT, USA. [11]Present address: Department of Chemical Engineering, Cooper Union, New York City, NY, USA. [12]Present address: Department of Cellular & Molecular Physiology, Yale School of Medicine, New Haven, CT, USA. [13]These authors contributed equally: Andreas Müller, Nikolai Klena, Gaia Pigino, Michele Solimena. ✉e-mail: andreas.mueller1@tu-dresden.de; gaia.pigino@fht.org; michele.solimena@tu-dresden.de

leads to abnormal beta cell polarity[18]. Perturbation of the expression of ciliary genes has been linked to the development of diabetes mellitus[19,20]. Moreover, in a mouse model of type 2 diabetes ciliogenesis appears to be impaired[20]. A deeper understanding of this highly specialized signaling compartment in beta cells is especially desirable in view of its proposed involvement in the pathogenesis of diabetes[20–23], a common metabolic disorder resulting from impaired beta cell insulin secretion.

The majority of cell types in our body have one primary cilium which can act as a "cellular antenna" for the reception of paracrine signals, and cilia dysfunction has been linked to a number of diseases usually referred to as ciliopathies such as Bardet-Biedl-syndrome or polycystic kidney/liver disease[24,25]. Cilia are microtubule-based organelles that extend from the cell forming a specialized membrane compartment[26]. The underlying microtubule array - the axoneme - extends from the basal body, which is constructed out of nine microtubule triplets (A-, B- and C-tubule) that continue as doublets after the termination of the C-tubule on the distal end of the basal body before the transition zone. Furthermore, in many cell types, a distinct domain termed ciliary pocket is present at the base of the cilium[27,28]. Cilia can be roughly divided into motile cilia (present in airways, sperm tails, etc.) and non-motile cilia, referred to as primary cilia[26]. Motile cilia have a complex and stereotyped structure with 9 microtubule doublets (A- and B-tubule) organized in a 9-fold symmetry at the periphery of the axoneme and 2 singlet microtubules at the center $(9 + 2)$[29]. This $9 + 2$ microtubular structure is the frame to which the motility protein complexes, such as axonemal dyneins and radial spokes, bind to generate ciliary beating[30–34]. Diversely, primary cilia are non-motile and their microtubule structure typically follows a $9 + 0$ scheme with a ring of 9 microtubule doublets[35]. However, serial section electron microscopy (EM) data have suggested the breaking of the 9-fold symmetry in close proximity to the basal body[36,37] leading to a variable axoneme structure[38]. Recently, reconstructions of complete primary cilia from cultured kidney cells revealed substantial variations from the $9 + 0$ architecture, with the progressive loss of microtubule doublets, as well as the B-tubule termination as the axoneme progresses distally[39,40]. Furthermore, axonemal microtubules consistently display a distortion from a radial arrangement, to a more bundled organization, with one or more singlets or doublets moving away from the axoneme towards the center of the cilia lumen. These data indicate that the 9-fold symmetry is not as relevant for the signaling function of primary cilia, as it is for the motility function in motile cilia.

Recently, it has been postulated that beta cell primary cilia can actively move in response to glucose stimulation[41]. However, the structural basis for this phenomenon is currently unclear. Until now the ultrastructure of beta cell primary cilia and their axonemes has been investigated mainly by two-dimensional (2D) transmission electron microscopy (TEM)[9,13,41,42] as well as scanning electron microscopy (SEM)[11,43]. Some of these studies indicate a disorganization of the axoneme. However, the complete arrangement of the microtubules of the primary cilium cannot be extrapolated from 2D TEM and SEM images. Previously, beta cells have been imaged using volume electron microscopy (vEM) methods[6,8,12], but the detailed 3D organization of the axoneme has not yet been reconstructed. Ultimately, full 3D reconstructions of beta cell primary cilia are necessary to understand their role in islet signaling. In addition to gathering information about the structure of the axoneme, direct visualization of primary cilia in the islet is of direct interest to gain insight into ciliary involvement in islet function.

In this study, we have employed two vEM methods with 3D segmentation accompanied by ultrastructural expansion microscopy (U-ExM) to investigate the 3D structure and molecular composition of beta cell primary cilia in the context of isolated islets and pancreas tissue. We found that the axoneme of beta cell cilia in mice and humans is disorganized and lacks motility components. We observed that beta cell primary cilia are spatially confined either within long ciliary pockets or restricted by surrounding cells, including acinar cells and islet vasculature. We also discovered close interactions of primary cilia originating from beta or alpha cells with neighboring islet cells or their cilia. Finally, we found that islet cell primary cilia can connect to and even form axo-ciliary synapses with the cholinergic innervation of the pancreas. Our structural and molecular data enable a better understanding of the beta cell primary cilia as compartments for signal transduction within the islet, as well as with the exocrine tissue and the autonomic nervous system, contextualizing beta cell primary cilia as major connective hubs in islet function.

## Results

### The axoneme of mouse and human beta cell cilia follows a $9 + 0$ organization which is not maintained over the whole cilia length

To resolve the organization of beta cell primary cilia, it is pivotal to obtain high-resolution 3D imaging data of their complete structure. This has so far been achieved for cells cultured in monolayers by serial section electron tomography of chemically fixed kidney cells[39,40]. Additionally, Kiesel et al.[39]. provided molecular data by cryo-electron tomography (cryo-ET) of kidney cilia identifying actin filaments within the axonemal structure and EB1 proteins decorating the lattice of the A-microtubules. However, high-resolution 3D reconstructions of complete primary cilia and their axonemes within tissues are currently not available. This is due to their low abundance and slender morphology, which makes their targeting and localization in EM datasets a tedious task. To reconstruct mouse beta cell primary cilia in situ we reviewed our recently published data[8,12,44–46] as well as previously unpublished datasets obtained by focused ion beam scanning electron microscopy (FIB-SEM) of isolated mouse islets fixed by high-pressure freezing (HPF) for the presence of primary cilia. We could indeed find primary cilia characterized by the presence of a basal body, axoneme, and ciliary membrane (Fig. 1a, Supplementary Fig. 1b). The voxel size of 4 nm was sufficient to discriminate microtubule doublets and singlets and the FIB-SEM volumes were large enough to contain complete primary cilia (Fig. 1a). This allowed for their 3D reconstruction (Fig. 1c, Supplementary Movie 1) and assessment of their interaction with the neighboring tissue. Especially for this task, fixation with HPF enabled optimal preservation of the ultrastructure because it avoids shrinking of the plasma membranes, as usually observed after chemical fixation.

To obtain data on human beta cell primary cilia we processed chemically fixed pancreas specimens from pancreatectomized living donors[47,48] and performed serial section electron tomography (ssET). Before the acquisition of electron tomograms, the serial sections were screened for the presence of basal bodies and axonemes by TEM (Supplementary Fig. 1a). The corresponding regions were then imaged by ET, and consecutive tomograms were aligned and stitched together. ET allows for imaging at higher resolution compared to FIB-SEM, so we decided for a voxel size of 1.307 nm in order to better resolve the microtubules as well as densities of other proteins of the cilium. We could detect primary cilia in human beta cells and again clearly resolve microtubule doublets and singlets of the axoneme (Fig. 1b, Supplementary Movie 2). Out of vEM we mostly reconstructed the primary cilia of beta cells, which we identified by the morphology of their secretory granules (SGs), consisting of dense insulin cores surrounded by a translucent halo[49–51] (Supplementary Fig. 1c). We refer to these cilia as beta cell primary cilia, whereas cilia with unknown cell origin are referred to as islet cell primary cilia.

The length of primary cilia in vEM datasets of mouse islets and human pancreas was 4.2 μm and 2.7 μm, respectively. Segmentation of the microtubules of the axoneme of the vEM data and subsequent analysis revealed a strong deviation from the classical $9 + 0$ organization, which was similar to that previously observed in 3D reconstructions of other cell types[39,40] (Supplementary Movie 3). As expected, in both mouse and human beta cell primary cilia the C-tubules of the 9 triplets of the basal body terminated before the so-called transition

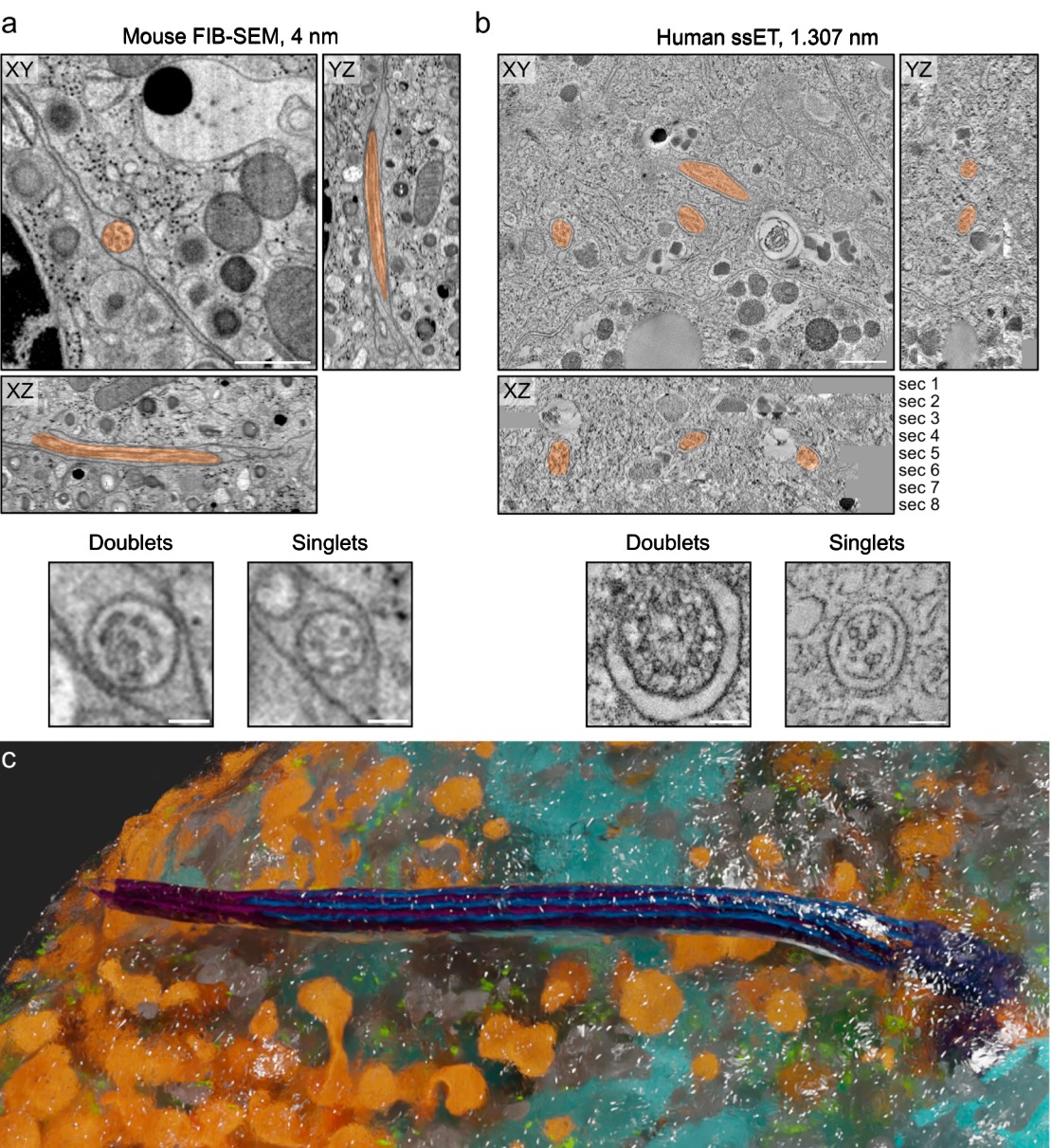

**Fig. 1 | vEM resolves mouse and human beta cell primary cilia ultrastructure.** FIB-SEM of one isolated mouse islet (**a**) and ssET comprising 8 consecutive sections of pancreas tissue of one living human donor (**b**). Primary cilia are marked in transparent orange. Both imaging modalities resolve the organization of the axoneme with microtubule doublets and singlets. Scale bars overview: 500 nm, scale bars axonemes: 100 nm. **c** 3D rendering of a mouse beta cell and its primary cilium with microtubules (blue, purple), mitochondria (light blue), insulin SGs (orange), endoplasmic reticulum (gray), ribosomes (green), and plasma membrane (transparent with white edges).

zone whereas A- and B-tubules continued more distally as 9 doublets (Fig. 2a, b). The termination of the C-tubules was better visible in the ssET data (Fig. 2b) due to the higher resolution. However, the axonemal doublets/singlets were clearly resolved both in FIB-SEM and ssET volumes. Over the length of the cilium some of the doublets were displaced towards the center of the cilium breaking axoneme symmetry and making its structure appear even more disorganized (Fig. 2b–d). During this transition, one or two doublets passed the center of the cilium getting closer to the opposite side (Fig. 2c, d). The maximum distance of these doublets to the cilia membrane was approximately 100 nm whereas doublets that stayed close to the membrane had a distance of approximately 40 nm to it.

In all reconstructed cilia the microtubule doublets were not present along the whole length of the organelle cilia, with the distal part of the axoneme comprising mostly single microtubules. vEM also

enabled the precise measurement of the lengths of A- and B-tubules. In the mouse cilium A-tubules had a mean length of $4.57 \pm 0.14\,\mu m$, and B-tubules a mean length of $3.03 \pm 1.01\,\mu m$ (Fig. 2e). In the fully reconstructed human beta cell primary cilium the length distribution was much more variable, with A-tubules measuring $2.05 \pm 1.01\,\mu m$ and B-tubules $1.20 \pm 0.65\,\mu m$ (Fig. 2e). In most cases the B-tubules terminated much before the A-tubules (Fig. 2a, c, d, Supplementary Movie 3), although we could observe cases in human beta cells where the opposite occurred (Supplementary Fig. 2). Furthermore, A- and B-tubules were slightly tortuous (Fig. 2f).

We carefully inspected the structure of axonemes, along microtubule doublets and singlets, in our vEM data for the presence of ciliary motility components, such as axonemal dyneins. We could not detect repetitive electron-dense structures compatible with the presence of motility components.

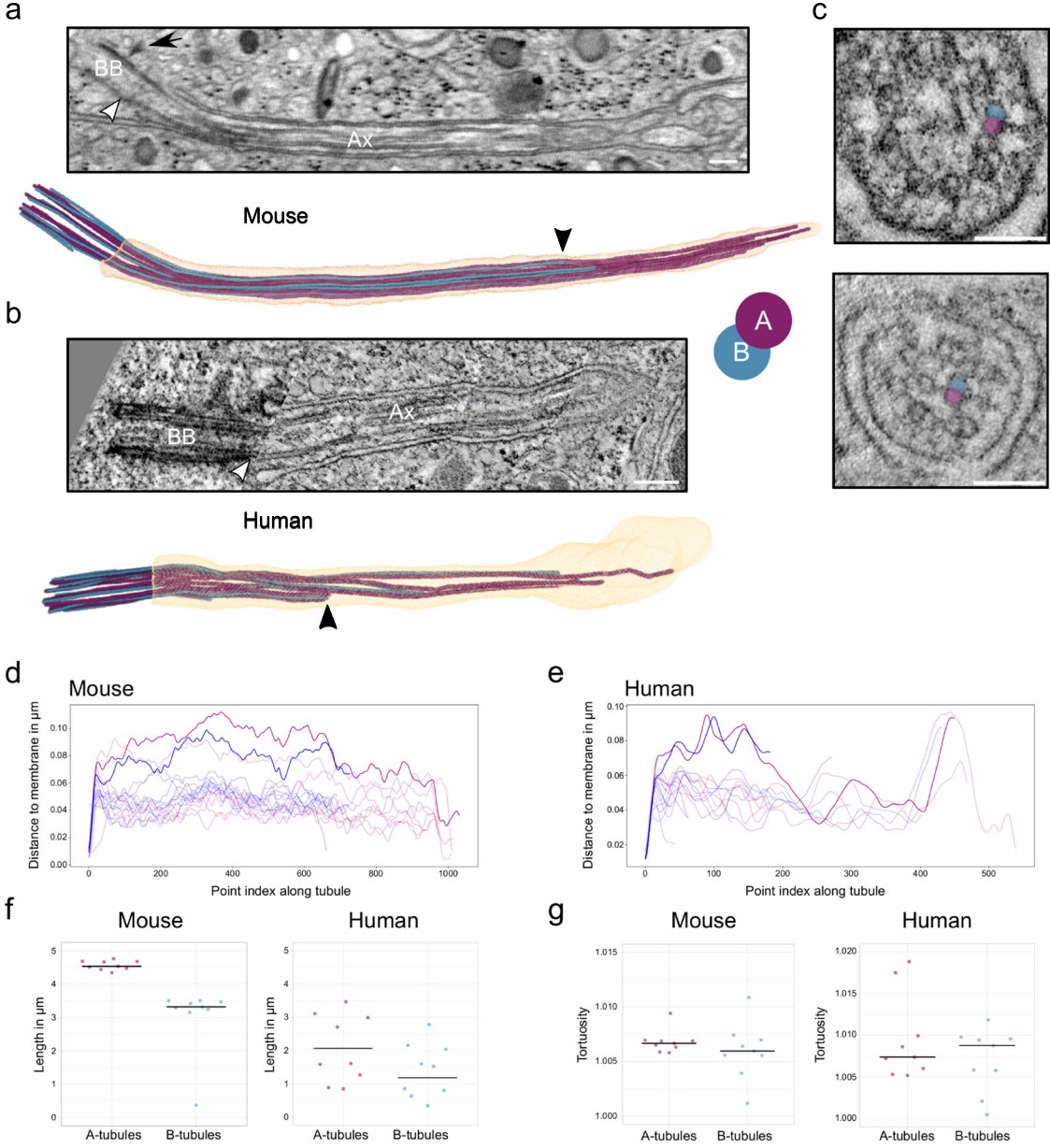

**Fig. 2 | The axoneme structure of primary cilia of mouse and human beta cells.**
**a** FIB-SEM slice through one mouse beta cell primary cilium. White arrowhead
indicates the termination of the basal body, black arrow indicates a distal appen-
dage. BB: basal body, Ax: axoneme. Scale bar: 200 nm. Below is the corresponding
3D rendering with ciliary membrane (light orange), A-tubules (purple), and
B-tubules (blue). The black arrowhead points to the termination of a B-tubule.
**b** ssET slice through one human beta cell primary cilium. The white arrowhead
indicates the termination of the basal body. BB basal body, Ax axoneme. Scale bar:
200 nm. The 3D rendering shows a human beta cell primary cilium from ssET
volumes. The black arrowhead points to the termination of a B-tubule.
**c** Displacement of microtubule doublets to the center of one human beta cell
cilium. A- and B-tubules of the example doublet are highlighted in purple (A) and
blue (B). Scale bar: 100 nm. **d** Distance of the microtubules of the mouse beta cell
primary cilium to the cilium membrane. Doublets with a major transition are
highlighted. **e** Distance of the microtubules of the human beta cell primary cilium to
the cilium membrane. Doublets with a major transition are highlighted. **f** Length
distributions of A- and B-tubules in respective segmentations of one mouse and one
human beta cell primary cilium. **g** Tortuosity of A- and B-tubules in respective
segmentations of one mouse and one human beta cell primary cilium. Source data
are provided as a Source Data file.

## U-ExM reveals the absence of functional motility complexes in beta cell primary cilia

The disorganization of the 9 + 0 structure of the beta cell ciliary axo-
neme combined with the lack of obvious dynein arms along the axo-
nemal length (Fig. 3b) is in contrast with recent reports of ciliary
motility components localizing to beta cell cilia, as well as active
motility in beta cells[41]. To address this point, we employed and opti-
mized ultrastructural expansion microscopy (U-ExM)[52] for imaging
primary cilia and putative ciliary components in mouse pancreas tissue
sections. This method allows for the crosslinking of a biological

specimen to a swellable hydropolymer, for its physical expansion and
super-resolution imaging. U-ExM has previously enabled the high-
resolution fluorescence imaging of centrioles[52] and intraflagellar
transport (IFT) trains within motile cilia[53] and connecting cilia in mouse
retina[54]. Furthermore, we validated the isotropy in the U-ExM of pan-
creatic sections by analysis nucleus cross-sectional (NCS) area in beta
cells pre and post-expansion (Supplementary Fig. 3a–c), as well as the
length and width dimensions of the daughter centriole by EM and
U-ExM (Supplementary Fig. 3d–f). We found no statistical difference in
the expansion factor corrected NCS area, or daughter centriole

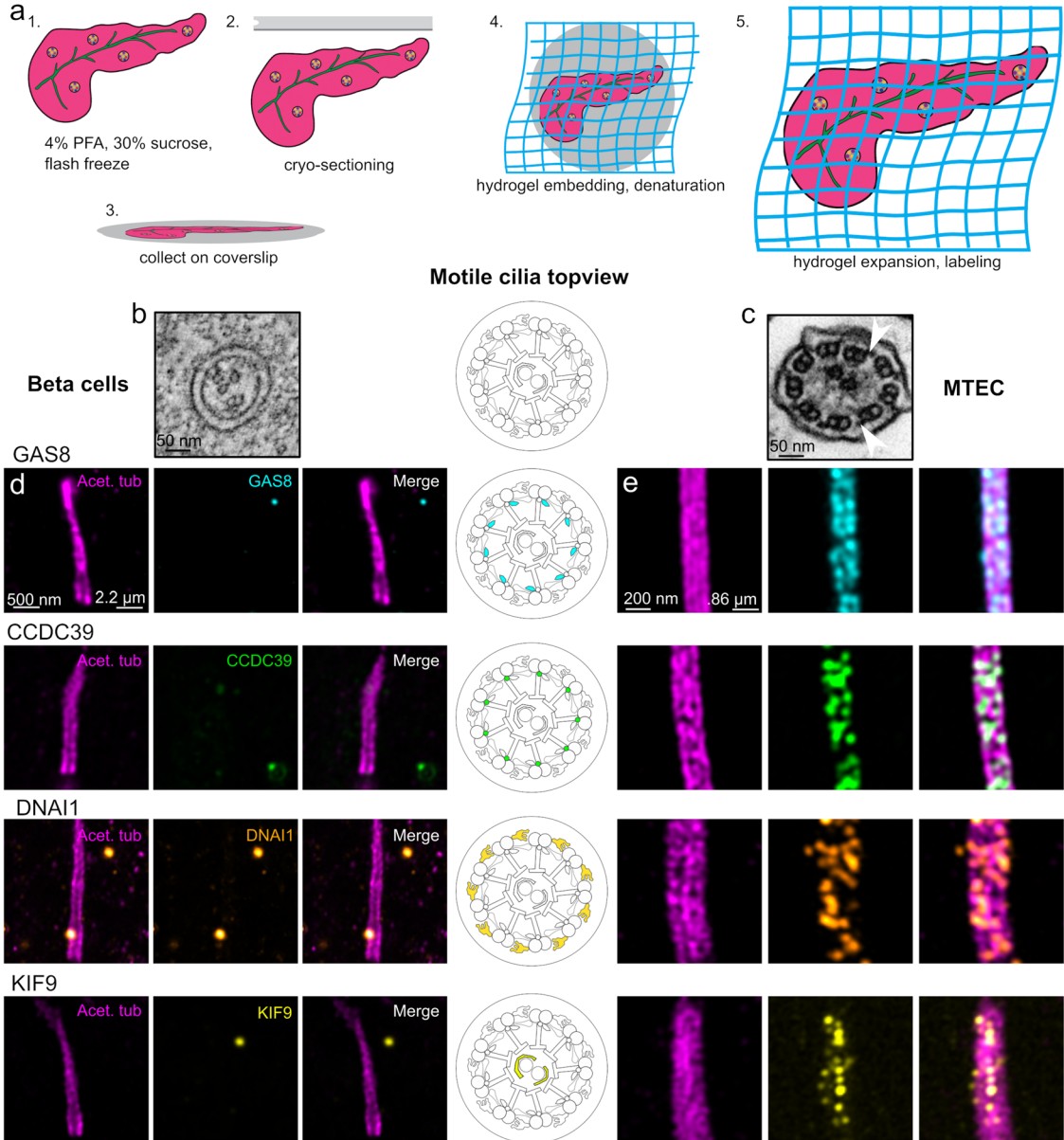

**Fig. 3 | Motility components do not localize to primary cilia of mouse beta cells.** **a** Schematic overview of pancreas U-ExM, including fixation, sectioning, collection, and expansion procedure. **b** Cross-sectional view of a beta cell primary cilium showing incomplete 9 + 0 organization. Scale bar: 50 nm. **c** Cross-sectional view of isolated mouse tracheal epithelium, displaying classic 9 + 2 organization. White arrows denote example outer dynein arms. Scale bar: 50 nm. **d** U-ExM images of beta cell cilia. Scale bar left, expansion factor corrected: 500 nm. Scale bar right, true scale: 2.2 μm. **e** U-ExM images of isolated mouse tracheal epithelium cilia (MTEC). Scale bar left, expansion factor corrected: 200 nm. Scale bar right, true scale, 0.86 μm. Cartoon indicates motility component localization in motile cilia. Component localization is determined by acetylated tubulin (magenta), GAS8 (cyan), CCDC39 (green), DNAI1 (hot orange), and KIF9 (yellow). U-ExM data were obtained from 2 individual mice.

dimensions, indicating general isotropy in the expansion of pancreas sections. U-ExM is therefore ideally suited to collect high-volume quantitative and molecular information on beta cell primary cilia and to localize proteins at the sub-organelle level.

We performed U-ExM on harvested adult pancreas, both by expanding the whole pancreas, and by collecting sections for expansion (Fig. 3a). We analyzed beta cell primary cilia for the presence of ciliary motility components, including GAS8 of the dynein-regulatory complex; CCDC39, which is part of the filamentous "ciliary ruler"; DNAI1, a component of the outer dynein arms; and KIF9, a central pair apparatus protein. As controls, we used isolated mouse airway cilia, which typically comprise these motility components. In 200 beta cell cilia, we could not detect GAS8, CCDC39, DNAI1, and KIF9 (Fig. 3d). As

expected, we observed complete decoration of the motility components along the length of the axoneme of mouse airway cilia (Fig. 3e), with the central pair marker KIF9 localizing along the central pair, GAS8 and CCDC39 directly on microtubules, and DNAI1 displaying a more exterior localization. These findings lead us to conclude that beta cell cilia are not actively motile.

## U-ExM enables quantitative molecular imaging of primary cilia in beta cells and pancreas tissue

We then set out to investigate how beta cells and their primary cilia interact with surrounding cells in the pancreas tissue. To this end, we further utilized U-ExM with fluorophore-conjugated NHS ester dyes, which bind to the amine groups of protein chains and allow for robust

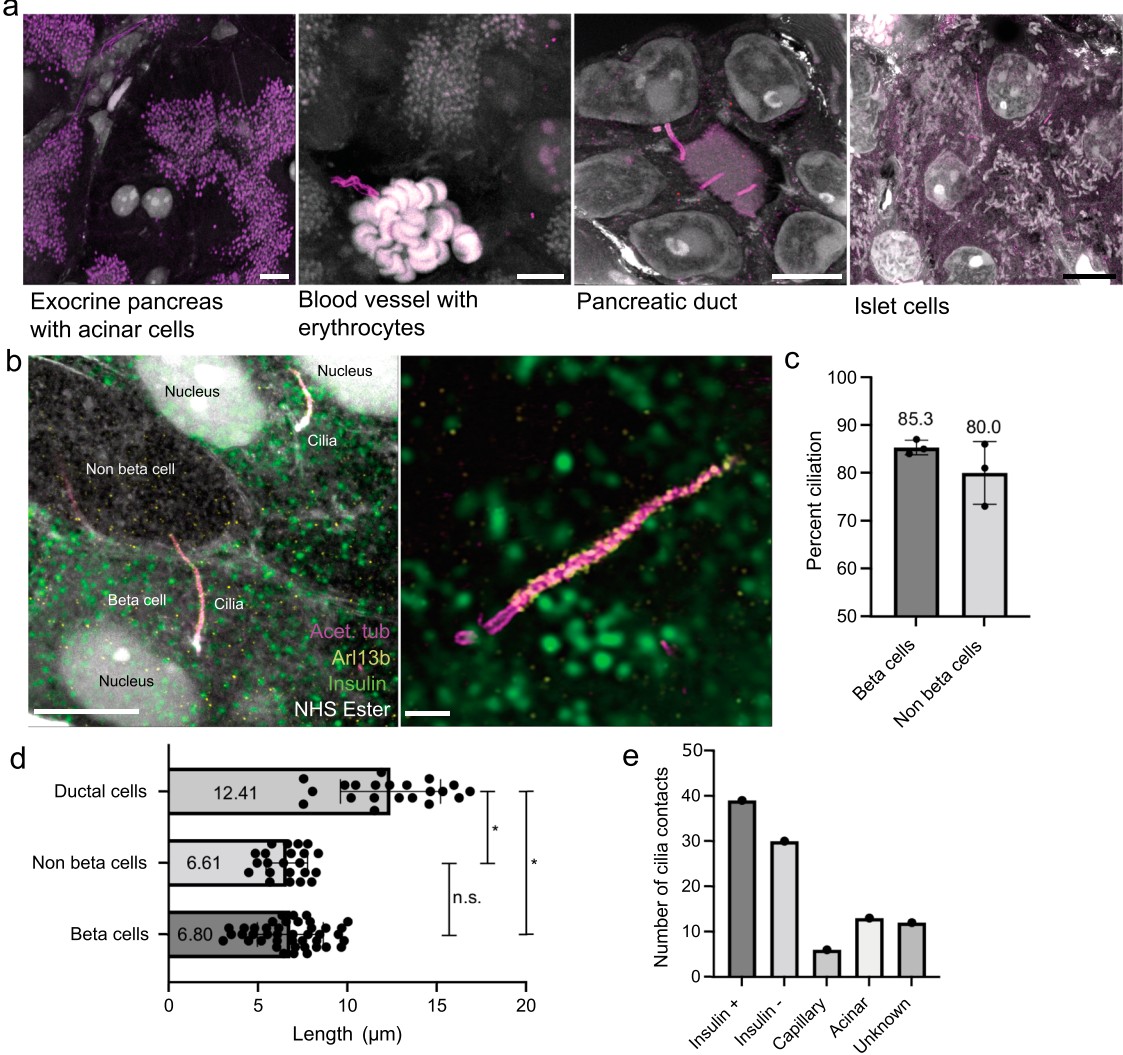

**Fig. 4 | U-ExM of mouse pancreas reveals cilia and cell type identity.**
**a** Representative confocal images of expanded pancreatic tissues labeled with antibodies for acetylated tubulin (purple) and with NHS ester for total protein (white). Scale bars, 10 μm. **b** Higher-magnification images of ciliated islet cells, including beta cells, stained for insulin (green), acetylated tubulin (magenta), Arl13b (yellow), and NHS ester. Scale bar, left: 5 μm. Scale bar, right: 1 μm.
**c** Quantification of percent ciliation in beta cells and non-beta cells within 3 islets from 3 different mice. 50 cells per islet were quantified. Mean ciliation percentage beta cells 85.3 % ± 1.528, non-beta cells, 80.0 % ± 6.557. Error is reported in standard deviation. **d** 3D length measurements of the cilium of ductal cilia, non-beta cells, and beta cells. Mean length: ductal cilia, 12.41 μm ± 2.8, non-beta cells 6.610 μm ± 1.158, beta cells 6.804 μm ± 1.841. Error is reported in standard deviation. Statistical significance is measured by unpaired, two-tailed, $t$-test. Ductal cells - beta cells: $p = 1.11 \times 10^{-13}$, ductal cells - non-beta cells: $p = 3.44 \times 10^{-11}$, beta cells - non-beta cells: $p = 0.6546$. $n = 42$ beta cells, 22 non-beta cells, 21 ductal cells from 2 pancreata. **e** Number of contacts of beta cell cilia to beta- and non-beta islet cells, as well as endothelial, acinar cells, and unknown cells. Source data are provided as a Source Data file.

fluorescent signals. With this approach, we could determine cell types with histology-like precision. Acinar cells were readily identifiable owing to the dark staining of zymogen granules by NHS ester and surprisingly, acetylated tubulin (Fig. 4a). Capillaries were detected by the presence of erythrocytes (Fig. 4a), and the canal-like architecture of ducts was also easily distinguishable (Fig. 4a). Islets were initially identified as large clusters without zymogen granules (Fig. 4a), and confirmed by insulin staining of the beta cells (Fig. 4b). To investigate potential differences in ciliary frequency, length, or appearance, we stained sectioned pancreas with antibodies against acetylated tubulin and Arl13b, both highly enriched in the cilium (Fig. 4b). Similar to previous reports[55], we found that 80–90% of islets cells were ciliated, including 85% of beta cells (Fig. 4c). Primary cilia in the U-ExM volumes had diverse shapes from straight to very bent (Supplementary Fig. 4). We further characterized pancreatic cilia by measuring ciliary length in a semi-automated manner. The cilia of pancreatic ducts were significantly longer (12.41 μm) than the islet cell cilia (approximately

6.8 μm) (Fig. 4d). We did not detect appreciable differences in the length of the cilia of the beta cells and other islet cell types (Fig. 4d).

Previous studies reported the presence of two primary cilia in a few beta cells[11,13]. We made similar observations in mouse pancreas tissue by U-ExM and found beta cells with two primary cilia (Fig. 5a). In all U-ExM images of mouse beta cell primary cilia we could observe a basal body and a daughter centriole for each cilium in double-ciliated cells, indicating that cilia were not originating from daughter centrioles in mouse beta cells. The basal bodies had variable distances to each other but the axonemes of both cilia were frequently found in close proximity to each other (Supplementary Fig. 5). Furthermore, in FIB-SEM data of mouse islets, we observed beta cells with two primary cilia (Fig. 5b, c). Here, we could discriminate between cilia sharing one ciliary pocket (Fig. 5b) and multiple cilia in separate pockets (Fig. 5c). In the latter case the cilia had very long ciliary pockets and left the cell on opposite sides, meeting other cilia as described in the following paragraphs (Supplementary Fig. 9). Notably, we did not detect beta cells with three cilia in our mouse datasets.

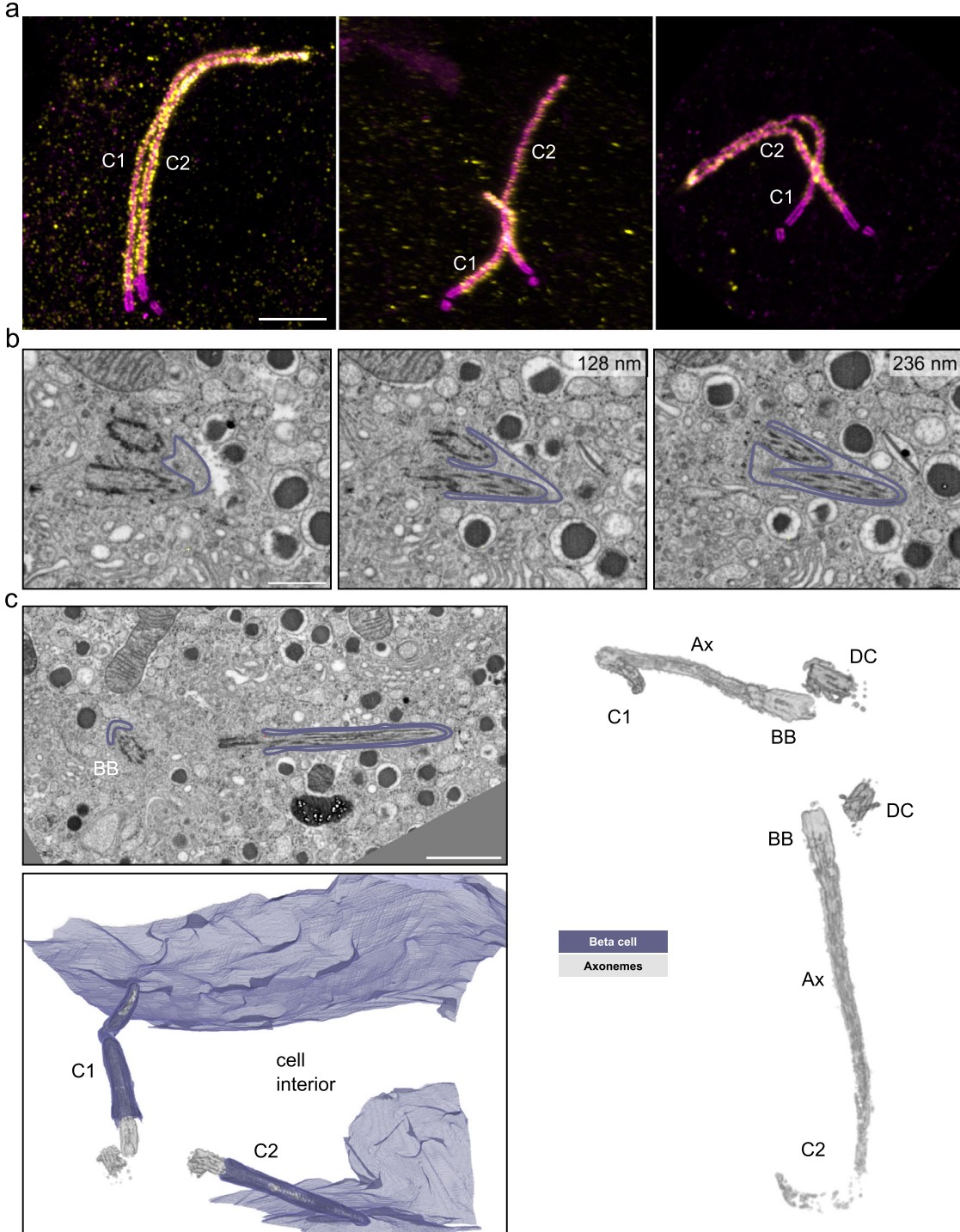

**Fig. 5 | Multiple cilia in beta cells. a** Expanded multiple islet cell cilia of one mouse pancreas stained by acetylated tubulin (magenta) and Arl13b (yellow). Scale bar: 1 μm. **b** Single FIB-SEM slices of one mouse pancreatic islet showing a beta cell containing two primary cilia sharing one ciliary pocket. The membrane is outlined in purple. Z-distances from the first image are indicated in the upper right of the following slices. Scale bar: 500 nm. **c** FIB-SEM data of one mouse pancreatic islet with a beta cell containing two primary cilia (C1 and C2) with distinct ciliary pockets. The cilia are pointing in opposite directions. The raw FIB-SEM slice shows the two cilia with basal bodies (BB) and membranes outlined in purple. Scale bar: 1 μm. The 3D rendering shows the plasma membrane in purple and the cilia microtubule structures in gray. The rendering on the right shows basal bodies (BB), daughter centrioles (DC), and axonemes (Ax) in gray.

While imaging islets by U-ExM, we frequently observed the cilia of beta cells making physical contact with other cells, including other beta cells, other islet cell types, and surprisingly, acinar and endothelial cells (Fig. 4e, Supplementary Fig. 7a). To investigate the ultrastructure of these interactions we used several EM methods, as described in the following paragraphs.

**Primary cilia are spatially restricted within long ciliary pockets and by neighboring cells**

Complementary to the quantitative data of U-ExM, EM of beta cell primary cilia deep within isolated islets and pancreas tissue allowed us to assess their ultrastructural features in situ. Previous SEM imaging[11] and our own SEM data of intact isolated islets only allow for imaging of

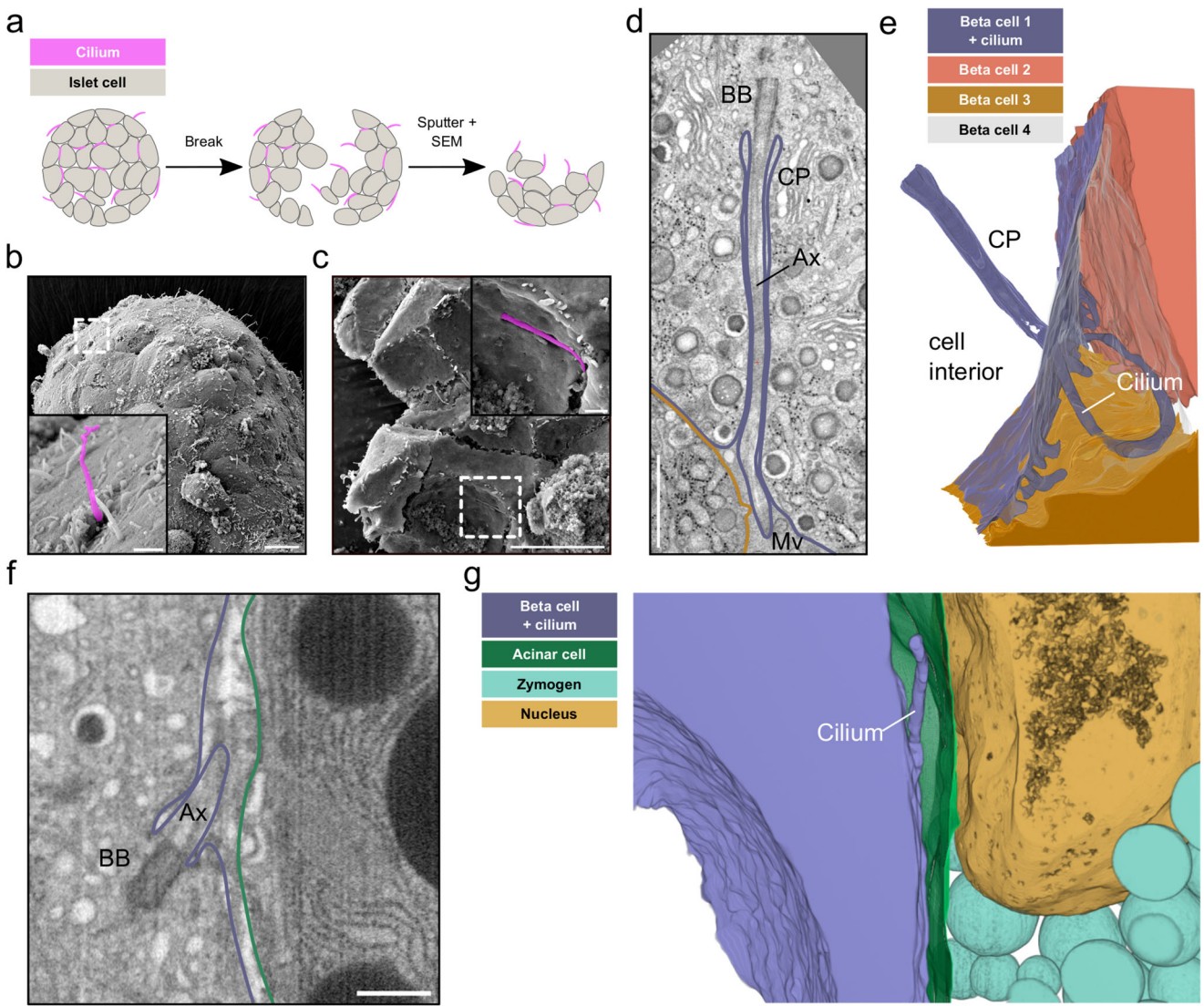

**Fig. 6 | Spatial restriction through ciliary pockets and neighboring cells.**
**a** Schematic workflow of breaking dried islets for SEM. **b** SEM image of the surface of one isolated mouse islet with islet cell primary cilia projecting into the extracellular space. The inset shows a primary cilium highlighted in purple. Scale bar overview images: 10 μm, insets: 1 μm. **c** SEM image of one broken isolated islet with a primary cilium restricted by the ciliary pocket. The inset shows a magnified view with the primary cilium highlighted in purple. Scale bar overview images: 10 μm, insets: 1 μm. **d** A single slice of a FIB-SEM volume of one mouse pancreatic islet with a beta cell primary cilium with a long ciliary pocket (CP). The cilium extends

towards the outside where it is surrounded by neighboring cells and microvilli (Mv). Ax: Axoneme, BB: Basal body. Scale bar: 1 μm. **e** 3D rendering of the segmentation shows the long ciliary pocket and the cilium (purple) surrounded by neighboring beta cells (orange, sand, gray). **f** A single slice of a FIB-SEM volume of one mouse pancreatic islet with a primary cilium of a beta cell on the edge of the islet close to the exocrine tissue. Scale bar: 500 nm. **g** Segmentation showing the beta cell primary cilium (purple) projecting into the extracellular space close to an acinar cell (green) with a nucleus (sand) and zymogen granules (light blue).

their peripheral surfaces. In such cases, the cilia appeared as structures extending into an empty extracellular space (Fig. 6b, Supplementary Fig. 6a), due to the collagenase-mediated digestion of the surrounding exocrine tissue and extracellular matrix for the isolation of the islets. However, even by SEM, it is possible to address a more natural tissue context by "breaking" the islets after critical point drying (Fig. 6a), as the fracture usually happens along the membranes of their cells (Supplementary Fig. 6b). This enabled us to visualize islet cell cilia with narrow ciliary pockets and in close contact to neighboring cells (Fig. 6c).

To further investigate the spatial restriction of primary cilia we segmented the cilia and their neighboring cells in our vEM volumes. We could detect a great heterogeneity in the length of the ciliary pockets, which could be several micrometers long (Fig. 6d), but in some cases extremely short and almost not detectable, such as the

mouse beta cell primary cilium in Fig. 1c. The ciliary pockets were in general very narrow, leaving only a few nm between the cilia membrane and the plasma membrane of the respective beta cell (Fig. 6d). Outside of the ciliary pocket, the cilia were mostly restricted in the narrow extracellular space between adjacent cells (Fig. 6d, e, Supplementary Movie 4, Supplementary Movie 1). Depending on their curvature, cilia were found in the space between 2 (see Fig. 1a, Supplementary Movie 1) or multiple islet cells (Fig. 5d, e, Supplementary Movie 4).

U-ExM had revealed a small percentage of beta cell cilia in close proximity to acinar cells and blood vessels. However, our FIB-SEM datasets did not contain these tissues (due to islet isolation) and our ssET of human pancreas data could only provide a limited z-depth and a small field of view. We therefore screened a public FIB-SEM dataset of mouse pancreas[56] including islets, acinar cells, and blood vessels for

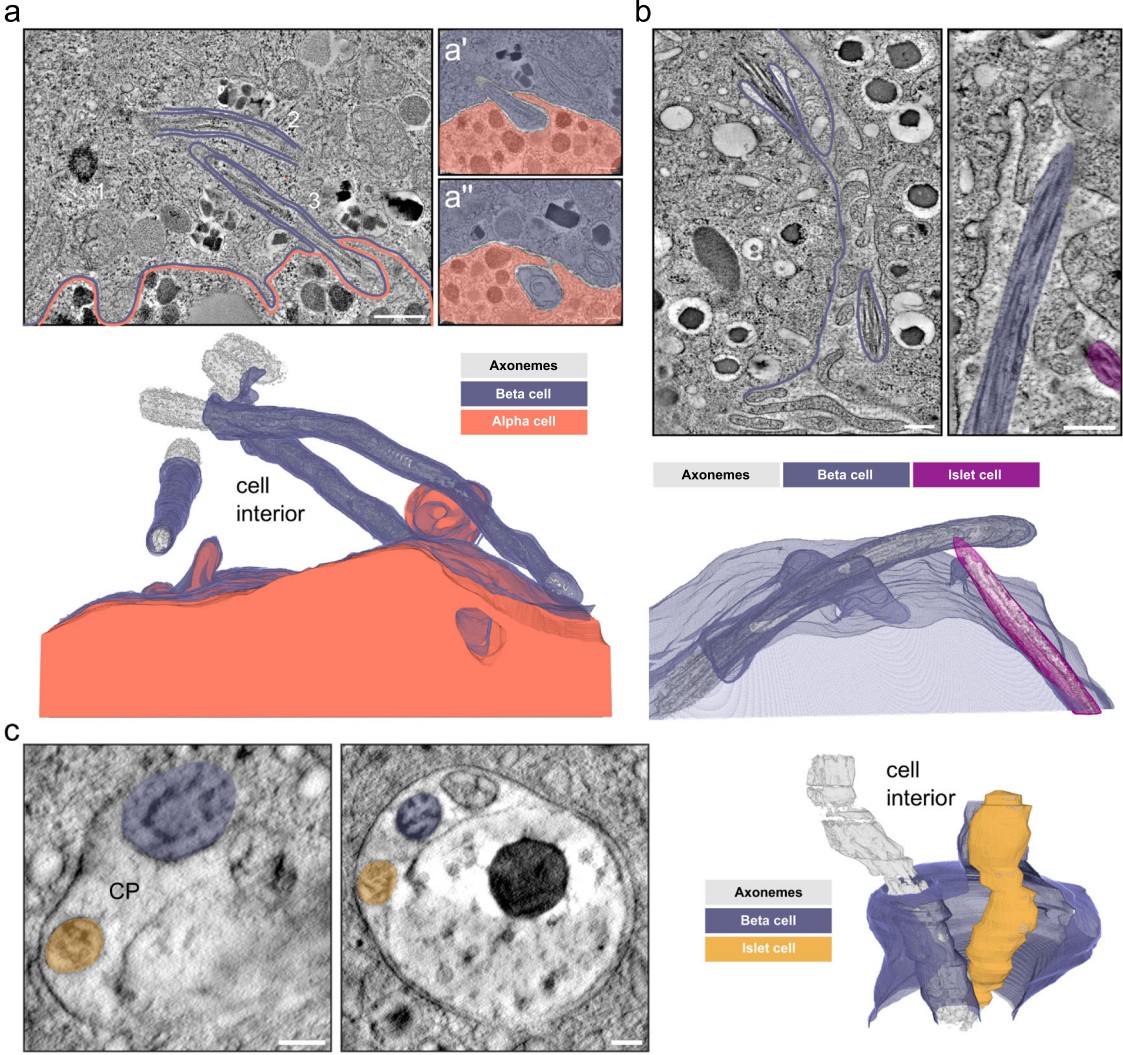

**Fig. 7 | Cilia-cell and cilia-cilia interaction. a** Single tomographic slices of primary cilia in human islets from one donor. There are 3 cilia originating in a beta cell with the basal body of one cilium (1) and the axonemes of 2 other cilia visible in the slice (2 and 3, purple). The neighboring alpha cell is outlined in orange. In magnified views of distant slices through the tomogram (a' and a") the end of cilium 2 is visible protruding and extending into a neighboring alpha cell (orange). Axonemes, basal bodies, and centrioles are rendered in gray. Scale bars: 500 nm and 100 nm. **b** Single tomographic slices of mouse beta cells of one isolated mouse pancreatic islet with 2 primary cilia (purple and magenta) originating from a beta cell (purple) and another islet cell (magenta) meeting in the space between cells. The side view shows the axonemes of the cilia in close proximity. The 3D view shows the segmentation of ciliary and plasma membranes together with the axonemes. Scale bars: 200 nm. **c** Tomographic slices of a ssET volume of one mouse pancreatic islet showing a large ciliary pocket shared by a primary cilium originating from a beta cell (purple) and a primary cilium of unknown origin (yellow). The 3D rendering shows the interaction of the two cilia. Axonemes are in gray. Scale bars: 100 nm.

the presence of primary cilia at the edge of the islets or close to the vasculature. We found beta cell cilia at the interface between the islet and exocrine tissue (Fig. 6f). The voxel size of these datasets of 8 nm was not sufficient to fully reconstruct the axonemes. However, the microtubule structure of these cilia resembled the disorganized 9 + 0 structure described above. 3D segmentation revealed that these cilia, similar to the ones deeper within the islet, were spatially restricted by either exocrine cells (Fig. 6f, g) and/or the extracellular matrix. In some cases we could also observe beta cell primary cilia by U-ExM and vEM in close proximity to blood vessels, projecting along their endothelial cells (Supplementary Fig. 7).

**Primary cilia closely interact with neighboring islet cells and cilia**

In our vEM volumes as well as in the U-ExM images we observed different modes of cilia connectivity: primary cilia contacting neighboring islet cells, and cilium-cilium contacts between cilia originating from distinct islet cells. It seems that contacts with neighboring islet cells

were mostly the result of the spatial restriction within the islet: once the cilium leaves the pocket and enters the extracellular space it is immediately surrounded by islet cells. Depending on the shape of the cilium we observed interaction with one or more islet cells. For instance, the highly curved cilium shown in Fig. 6d touched three neighboring cells. When investigating these interactions at high resolution we did not detect direct contacts of ciliary membranes with plasma membranes, as there was always a small gap of a few nanometers between them without any electron-dense regions characteristic of gap junctions. Furthermore, we observed cilia deepening into neighboring cells surrounded by their strongly deformed plasma membrane (Fig. 7a). This resulted in tunnel-like structures in the neighboring cells, generating synapse-like connections. The beta cell in this volume had three primary cilia (Fig. 7a). The cilia appeared to have similar structural features as those from single-ciliated beta cells with the same disorganized axonemes. We also observed an alpha cell cilium in the FIB-SEM dataset of mouse pancreas[56] projecting into a neighboring alpha cell ending close to its nucleus (Supplementary

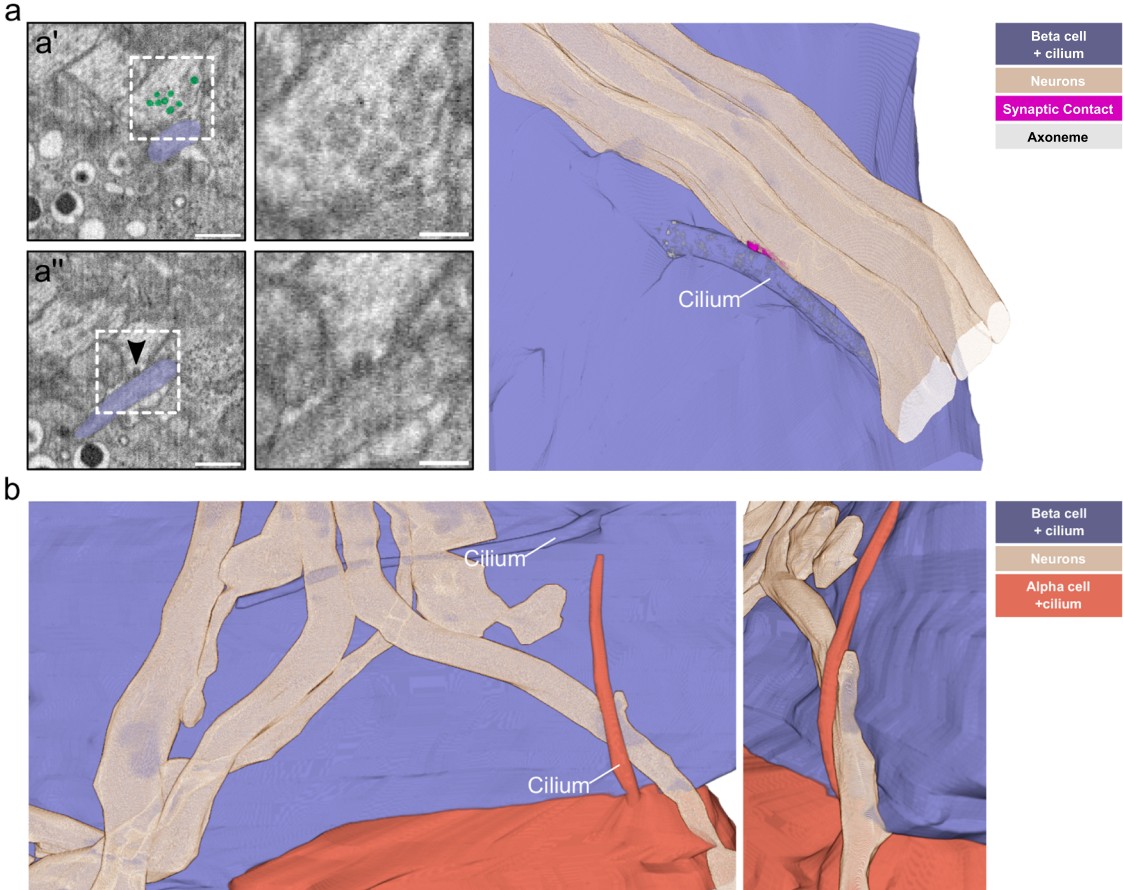

**Fig. 8 | Islet cell primary cilia-neuron connections. a** A single slice of a FIB-SEM volume of one P7 mouse pancreas sample with an axon and a beta cell with a primary cilium (purple) in contact with each other. Scale bar: 1 μm. In a', synaptic vesicles close to the cilium-neuron contact are highlighted in green. Scale bar: 500 nm. The boxed area is shown magnified in the right panel without highlighting the vesicles. Scale bar: 100 nm. The arrowhead in a" points to a synaptic vesicle likely undergoing exocytosis towards the cilium. Scale bar: 500 nm. The magnified image in the right panel shows the area of the fusion event. Scale bar: 100 nm. The 3D rendering shows the contact of a primary cilium (purple) with one of the axons (beige), including a segmentation of the synaptic density (magenta). **b** 3D rendering of primary cilia from a beta (purple) and an alpha cell (orange) in contact with the same axon (beige). The smaller image shows a tilted magnified view of the alpha cell cilium-axon interaction.

Fig. 8a, b). In turn, the primary cilium of this second alpha cell protruded into a third adjacent alpha cell (Supplementary Fig. 8c, d).

Additionally, we observed cilium-cilium contacts between distinct islet cells in the extracellular space (Fig. 7b). In most cases cilia from two distinct cells projected into the same region surrounded by microvilli, with their tips in close proximity to each other. We also observed cilia that, protruding from the opposite sides of a double-ciliated beta cell, contacted other cilia from adjacent cells (Supplementary Fig. 9). Moreover, in one instance we even found cilia from distinct islet cells meeting in the same ciliary pocket of a beta cell (Fig. 7c). This ciliary pocket was relatively wide compared to the other pockets we observed and it also contained large vesicular structures.

### Islet cell primary cilia connect to the pancreatic innervation

Research has focused so far on the role of islet cell cilia in paracrine signaling between endocrine cells. Recently, however, primary cilia have been shown to form synapses with neurons in brain tissue[57,58]. While screening the aforementioned 8 nm isotropic FIB-SEM dataset of P7 mouse pancreas tissue[56] we found several nerve fibers contacting the islets (Supplementary Fig. 10a). These fibers showed classical structural features of neuronal axons, being very long, containing cables of parallel microtubules and synaptic vesicles (Supplementary Fig. 10a). They usually reached the edge of the islet and occasionally split into few fibers when transitioning deeper into the islet. They contacted the plasma membranes of islet cells including alpha, beta, and delta cells. Surprisingly, we detected primary cilia (as identified by the presence of basal body, daughter centriole, and axoneme) of beta and alpha cells projecting towards these nerve fibers and contacting them (Fig. 8a, b, Supplementary Fig. 10b, c, Supplementary Movie 5). On some of these contacts, we could observe structural features of presynapses on the side of the neurons: synaptic vesicles close to these contacts and possibly synaptic vesicle exocytosis (Fig. 8a', a", Supplementary Movie 5). Furthermore, the contacts seemed to be more electron-dense compared to cilia and neuron membranes alone. Overall, our observations closely resembled the structural features of axo-ciliary synapses[57]. We observed these contacts being formed by both beta and alpha cell cilia, as the axons traversed through the islet (Fig. 8b). In this case the primary cilia of a beta cell and the adjacent alpha cell connected with the same nerve fiber. In most cases, cilia contacted axons with their lateral side. However, in one case we could also observe the tip of an alpha cell cilium touching an axon (Supplementary Fig. 10b, c).

### Axonal varicosities contacting beta cell cilia are positive for synapsin 1 and vAChT

We next aimed to better characterize these putative axo-ciliary synapses by performing immunofluorescence stainings for presynaptic markers on sections of the adult mouse pancreas. We labeled

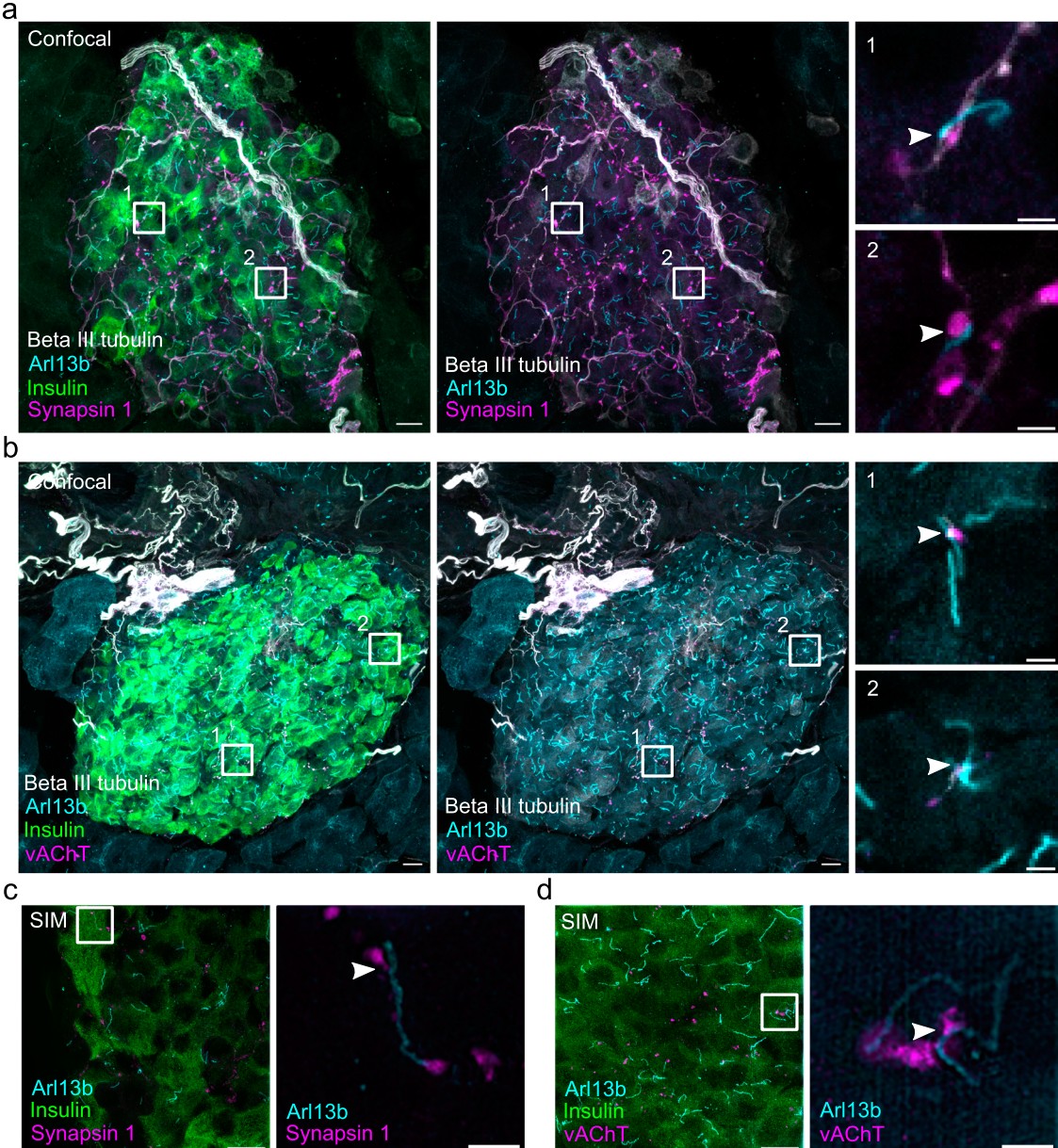

**Fig. 9 | Presence of synapsin 1 and vAChT at neuron-cilia contact sites.**
**a** Maximum intensity confocal images of a cryo section of mouse pancreas after immunofluorescence labeling for insulin (green), beta III tubulin (white), Arl13b (cyan), and synapsin 1 (magenta). The center image shows the same region without the insulin. Scale bars: 10 μm. Regions 1 and 2 show Arl13b in close proximity to beta III tubulin and synapsin 1 and single confocal slices of these regions are magnified on the right. Scale bars: 2 μm. **b** Maximum intensity confocal images of a cryo section of mouse pancreas after immunofluorescence for insulin (green), beta III tubulin (white), Arl13b (cyan), and vAChT (magenta). The center image shows the same region without the insulin. Scale bars: 10 μm. Regions 1 and 2 show Arl13b in close proximity to beta III tubulin and vAChT and single confocal slices of these regions are magnified on the right. Scale bars: 2 μm. **c** Maximum intensity SIM images of a cryo section of mouse pancreas after immunofluorescence labeling for insulin (green), Arl13b (cyan), and synapsin 1 (magenta). Scale bar: 10 μm. The magnified box shows a single SIM slice with Synapsin 1 signal in close proximity to Arl13b. Scale bar: 2 μm. **d** SIM image of a cryo section of mouse pancreas after immunofluorescence labeling for insulin (green), Arl13b (cyan), vAChT (magenta). Scale bar: 10 μm. The magnified box shows a single SIM slice with vAChT signal in close proximity to Arl13b. Scale bar: 2 μm. Data were obtained from 2 mouse pancreata. Source data are provided as a Source Data file.

neurons with an antibody for beta III tubulin to image their axons and identified beta cells by staining for insulin. Primary cilia were localized by immunolabeling for Arl13b. These stainings showed large nerves penetrating the islet and splitting partially into single axons which contacted islet cells (Fig. 9a). We found that 12.5 ± 2.8% of beta cell cilia were in contact with axons. Next, we wanted to find out if these contacts possess markers of functional synapses. Initially, we aimed to identify synapses by staining for the presynaptic marker synaptophysin 1, which has been used to characterize axo-ciliary synapses previously[57]. However, since synaptophysin 1 is also present in the

synaptic-like microvesicles of islet cells[59] these stainings did not allow for easy detection of axo-ciliary synapses, although we observed the signal of synaptophysin 1 colocalizing with beta III tubulin (Supplementary Fig. 11a).

We therefore used synapsin 1, a bona-fide marker for presynaptic terminals[60], which has been shown to be present on islet cell innervation[61] and is barely expressed in beta cells. Synapsin 1 colocalized with beta III tubulin at axonal varicosities characteristic for en passant synapses (Supplementary Fig. 11b). In co-stainings for Arl13b we could identify axonal varicosities positive for synapsin 1 in close

proximity to primary cilia (Fig. 9a). In these images we found that 70.8 ± 3.7% of contacts between beta cell primary cilia and axons were positive for Synapsin 1. In order to resolve these contacts better we turned to structured illumination microscopy (SIM) which also revealed beta cell primary cilia in close proximity of distinct synapsin 1 positive objects (Fig. 9c). Notably, the signals of beta III tubulin and synapsin 1 did not overlap with Arl13b indicating their localization to distinct compartments. These stainings therefore demonstrate the presence of presynaptic markers in the axons at beta cell axo-ciliary synapses.

Next, we aimed to identify the type of neurotransmitters involved in beta cell axo-ciliary signaling. Since acetylcholine plays a major role in the neuronal control of islet function[62] we searched for the presence of acetylcholine-positive synaptic vesicles in axons contacting primary cilia. Immunostaining for the vesicular acetylcholine transporter (vAChT), which is responsible for loading acetylcholine into synaptic vesicles, showed positive signals at most axonal varicosities together with synapsin 1 (Supplementary Fig. 11c). Likewise, we identified vAChT positive signals in neurons in close proximity to Arl13b (Fig. 9b) at the putative axo-ciliary synapse. In these stainings, 64.2 ± 27.7% of contacts between beta cell primary cilia and axons were positive for vAChT. As for synapsin 1, vAChT signal was present at the contact site of cilium and nerve and could be resolved at higher resolution by SIM (Fig. 9d). Notably, immunoreactivity for vAChT was also detected at the plasma membrane of beta cells, as visualized by costaining for the surface adhesion molecule (EPCAM) and insulin (Supplementary Fig. 12). These findings lead us to conclude that acetylcholine is present at the presynaptic site of beta cell axo-ciliary synapses.

## Discussion

We provide here an in-depth volumetric characterization of primary cilia within mouse and human tissue by applying vEM and U-ExM, thus resolving the structural organization of the axoneme and giving a comprehensive overview on cilia interaction and connectivity. High-resolution 3D data have so far only been available for primary cilia of monolayer cells facing the cell culture medium[39,40]. Our data allow us to resolve the structure of the axoneme as well as the spatial interaction of islet beta cell primary cilia in their native environment. We can show that the 9 + 0 axoneme is not fully maintained throughout the whole length of the cilium in mouse and human beta cell cilia. Together with neurons, glia, and cultured kidney cells, the present data uncovers another cell type in which a continuous 9 + 0 organization is the exception rather than the rule and strongly suggests that most primary cilia deviate from this organization. In the reconstructed human beta cell cilium the length of microtubules of the axoneme varied much more compared to the mouse beta cell cilium. This might be explained by the older age of the human beta cells we analyzed. However, these values will need to be corroborated by data from more cells in the future. Notably, the primary cilium has been shown to be among the oldest organelles in beta cells[63], which might explain its disorganization as cells age. We further show the absence of a central microtubule pair along the full length of the cilium. Instead, some of the outer microtubule doublets/singlets are displaced towards the center of the cilium, which explains previous incorrect observations of central pair microtubules in 2D sections[41]. While these data do not per se fully rule out the possibility of cilia movements, they indicate that the phenomena observed by Cho et al. cannot be mediated by the canonical microtubule organization of motile cilia. This conclusion is corroborated by the lack of density corresponding to dynein arms in primary cilia by vEM, as well as by the failure to detect with U-ExM the motility components GAS8, CCDC39, DNAI1, and KIF9 in any of the 200 beta cell cilia we examined. Notably, primary cilia have been shown to move by Brownian motion with the involvement of the actin cytoskeleton surrounding the basal body[64]. Imaging beta cell primary cilia in their in situ environment - isolated pancreatic islets but also

pancreatic tissue - enabled us to investigate their interactions within the native context. We found that beta cell cilia are spatially restricted by their own ciliary pockets and the neighboring cells, leaving only a few nanometers of freedom. This spatial restriction resulted in the formation of ciliary connections with one or more neighboring islet cells depending on the shape and curvature of the cilium. Furthermore, some cilia formed synaptic-like structures pinching neighboring cells. It is attractive to speculate that spatial and connection heterogeneity among beta cell cilia subserves different structural and signaling functions.

Cilia are known to transduce signaling pathways, such as WNT[65] and hedgehog[66] and it has been shown that WNT4 regulates the physiology of mouse adult beta cells[67]. Beta cell cilia have been shown to be involved in GABA signaling[68]. Furthermore, they express the insulin[19] and somatostatin[69,70] receptors, and various G protein-coupled receptors, including the type 2 diabetes-risk gene free fatty acid receptor 4 (FFAR4)[55]. If some of these pathways are especially prominent at cilia-islet cell connections remains to be investigated as the molecular factors involved in the establishment of these connections and the timing of their formation are still unknown.

Along this line, it is known that beta cells are heterogeneous and that so-called "hub" or "leader" beta cells play a specialized role in islet calcium signaling and propagation of insulin secretion[71,72]. Recently, hub cells have been shown to express a number of differentially regulated cilia genes relative to follower cells[73]. The correlation of fluorescently labeled hub cells with EM, U-ExM, or dynamic live cell imaging could allow for investigating their cilia structure in the future.

The frequently observed double-ciliated beta cells could also play a special role in islet connectivity. In each example of multiple ciliated beta cells, two complete copies of the centrosome were present, with the cilium emanating from the mother centriole, and two daughter centrioles present in the vicinity. This feature is in contrast to the multiple cilia occasionally observed in cultured cells, where a cilium emerges from the daughter centriole or bifurcates near the base. Cilia emerging from two distinct mother centrioles reflects a controlled and distinct process of centrosome duplication to allow the formation of two complete independent signaling entities.

Even more surprising to us was the observation of interactions between cilia originating from distinct types of islet cells. The contacts occur not only in the extracellular space but even in shared ciliary pockets. These phenomena could be the result of the special islet architecture[5,6], characterized by the dense packing of islet cells. Moreover, the destruction of beta cell cilia by deletion of *IFT88*[18] resulted in loss of beta cell polarity. Thus, our observations are unlikely to capture random events but reflect specific signaling events that attract cilia to each other. The molecular pathways and factors supporting these cilium-cilium interactions are largely unknown, but it has been shown that glycoprotein-mediated contacts establish long-term cilium-cilium interactions in MDCK cells[74]. Whether a similar type of mechanism supports cilium-cilium interactions within islet cells remains to be investigated. Additionally, cilia have been described to support cell connectivity in the brain[75,76] and might play a similar role within pancreatic islets. The evidence that beta cells can establish connections with neighboring cells through multiple cilia further support the idea of cilia being relevant for paracrine communication.

Recent publications investigating the role of primary cilia in adult human brain cortex[58] and mouse visual cortex[77] have revealed ultrastructural differences in the cilia pertaining to diverse cell types, specifically in the morphology of the ciliary pocket. Interestingly, the embedding of beta cell primary cilia deep into ciliary pockets closely resembles features observed in astrocytes[58,77]. These cells have a well-described role in maintaining connectivity with endothelial cells at the blood–brain barrier[78]. We speculate that physical contacts between beta cell cilia and endothelial cells, visible in our data, modulate their reciprocal signaling. Notably, we found beta cell cilia with either very

long or very short ciliary pockets. In neuronal tissue cilia with long pockets do not reach as many cells as cilia with short pockets[58,77]. However, cilia with long pockets have more surface area for potential autocrine signaling. Furthermore, the ciliary pocket is a hub for membrane trafficking, such as endocytosis[79], which could be enhanced in cilia with long pockets.

Finally, the observation of synapses between islet cell cilia and innervating axons points to signaling functions of primary cilia beyond the local islet context. To our knowledge, axo-ciliary synapses have only been described in the brain[57,58]. Thus, our reconstructions are the first to document these connections elsewhere in the body. The localization of synapsin 1 and vAChT on varicosities in close proximity to beta cell primary cilia indicates the presence of functional presynaptic terminals. Acetylcholine plays a major role in the autonomic regulation of islet cells and is involved in the cephalic phase of insulin secretion[62,80]. Beta cell primary cilia could therefore play a role in receiving cholinergic stimuli before an increase in blood glucose levels occurs. It has been proposed that the signaling from the primary cilium may have more direct access to the nucleus and thus can more easily modulate chromatin accessibility[57]. The downstream effects of axo-ciliary signaling on beta cell function still need to be elucidated.

Similar to axo-ciliary synapses in the brain[57], also in the islets we could not detect postsynaptic markers in the primary cilia contacting neurons. Furthermore, the presence of postsynaptic proteins in islet cells has only rarely been reported[81]. However, primary cilia were shown to share structural and functional similarities with dendritic spines[82,83]. This, together with the presence of numerous GCPRs on beta cell cilia[55,68,69], suggests that they can act per se as a postsynaptic compartment. Recent work further postulates that the close vicinity of cilia to synapses enables them to detect neurotransmitters[58,77]. This could also be the case for islet cell cilia touching axons without the formation of axo-ciliary synapses. Future work will be required to determine which ciliary-localized proteins are acting in a postsynapse capacity in the axo-ciliary synapse.

The variety of interactions we have investigated here - cilia-islet cell, cilia-cilia, and cilia-neuron contacts - point to a functional heterogeneity of beta cell primary cilia depending on the tissue/organelle they connect to. We propose that cilia may differ in the composition of their membrane depending on their localization in the islet. For example, somatostatin receptor 3 was not detected in all beta cell cilia[16]. Different cilia functions could also depend on the origin of the signals they receive.

In summary, we provide comprehensive structural insights into the role of primary cilia as a key compartment for islet cell connectivity and signaling. We provide data on beta cell cholinergic axo-ciliary synapses, which supposedly play a role in the integration of systemic neuronal inputs. Our work opens new research questions on the pathways for establishing cilia connectivity throughout the body and the molecular factors involved in ciliary signaling.

## Methods

### Islet isolation and culture
The isolation of islets of Langerhans of 9-week-old C57BL/6 mice was performed as previously described in ref. 84. Islets were cultured overnight in standard culture media (Roswell Park Memorial Institute 1640 [Gibco] with 10% FBS, 20 mM Hepes, and 100 U/ml each of penicillin and streptomycin) with 5.5 mM glucose. Before processing by HPF, the islets were incubated for 1 h in Krebs–Ringer buffer containing either 3.3 mM or 16.7 mM glucose. All animal experiments were conducted according to the guidelines of the Federation of European Laboratory Animal Science Associations (FELASA) and are covered by respective licenses for those experiments from the local authorities. Facilities for animal keeping and husbandry are certified and available with direct access at the Dresden campus (including facilities at Paul Langerhans Institute Dresden and Max Planck Institute for Molecular

Cell Biology and Genetics). Licenses for animal experiments are approved by the State Directorate Saxony under license number DD24.1-5131/450/6.

### Preparation of pancreatic specimens of living donors for EM
Pancreas specimens from a cohort of patients from the University Hospital Carl Gustav Carus described in refs. 47,48 were examined by a certified pathologist after resection. The study was conducted with the ethical approval of the Ethical Committee of the Technische Universität Dresden (Study No.: EK 151062008) including informed consent of the patients. The samples were cut into cubes with a side length of max. 1 mm and fixed with 4% glutaraldehyde in sodium phosphate buffer immediately after dissection. Samples were block-contrasted with 1% osmiumtetroxide followed by 1% uranylacetate. After dehydration, samples were embedded in epoxy resin (Embed 812, Science Services). Ultrathin sections were cut with a Leica U6 ultramicrotome (Leica Microsystems) and post-stained with uranylacetate and lead citrate. Grids were screened on a JEM 1400 Plus with a Jeol Ruby camera. The sample with the best preservation of ultrastructure (from a patient with impaired glucose tolerance) was used for cutting serial sections for ssET.

### High-pressure freezing and freeze substitution
Isolated islets were kept in culture overnight and frozen with a Leica EMpact 2 or EM ICE high-pressure freezer (Leica Microsystems). They were kept in liquid nitrogen until freeze substitution. Freeze substitution was performed according to a previously published standard contrast protocol[85] or according to the following high contrast protocol[8,86]: first, the samples were substituted in 2% osmiumtetroxide, 1% uranylacetate, 0.5% glutaraldehyde, 5% $H_2O$ (according to ref. 87) in acetone with 1% methanol at −90 °C for 24 h. The temperature was raised to 0 °C over 15 h followed by four washes with 100% acetone for 15 min each and an increase in temperature to 22 °C. Afterwards, the samples were incubated in 0.2% thiocarbohydrazide in 80% methanol at RT for 60 min followed by 6 × 10 min washes with 100% acetone. The specimens were stained with 2% osmiumtetroxide in acetone at RT for 60 min followed by incubation in 1% uranylacetate in acetone plus 10% methanol in the dark at RT for 60 min. After four washes in acetone for 15 min each, they were infiltrated with increasing concentrations of Durcupan or Epon resin in acetone followed by incubation in pure Durcupan and polymerization at 60 °C.

### Serial sectioning and electron tomography
Blocks were sectioned with a Leica UC6 ultramicrotome (Leica Microsystems), and 300 nm serial sections were put on slot grids containing a Formvar film. Sections were contrasted with 2% uranylacetate in methanol followed by 1% lead citrate in $H_2O$. Tilt series ranging from −63° to +63° were acquired with a F30 EM (Thermo Fisher Scientific) equipped with a Gatan US1000 camera (Gatan). The tomograms were reconstructed and joined with the IMOD software package[88].

### FIB-SEM
Multiple Durcupan-embedded isolated islets samples were each first mounted to the top of a 1 mm copper post which was in contact with the metal-stained sample for better charge dissipation. The region of interest (ROI) was identified by x-ray tomography data obtained with a Zeiss Versa XRM-510 and optical inspection under a microtome. These vertical sample posts were each trimmed to its defined ROI using a Leica UC7 ultramicrotome. This sample preparation methodology was developed for precise ROI targeting and improved image acquisition, as previously described[89]. Before FIB milling, a thin layer of conductive material of 10 nm gold followed by 100 nm carbon was coated on each sample with a Gatan 682 Precision Etching and Coating System with the following coating parameters: 6 keV, 200 nA on both argon gas

plasma sources, 10 rpm sample rotation with 45° tilt. After coating, each sample was imaged with a customized, enhanced FIB-SEM consisting of a Zeiss Capella FIB column mounted at 90° onto a Zeiss Merlin SEM as previously described[12,90]. Each block face was imaged by a 140 pA electron beam with 0.9 keV landing energy at 200 kHz. The x-y pixel size was set at 4 nm. A subsequently applied focused Ga+ beam of 15 nA at 30 keV strafed across the top surface and ablated away 4 nm of the surface. The newly exposed surface was then imaged again. The ablation–imaging cycle continued about once every 3–4 min for several days up to two weeks to complete the FIB-SEM imaging of one sample. The sequence of acquired images formed a raw imaged volume, followed by post-processing of image registration and alignment using a Scale Invariant Feature Transform-based algorithm. The actual z-step was estimated by the changes of SEM working distance and FIB milling position. The image stacks were rescaled to form $4 \times 4 \times 4$ nm isotropic voxels, which can be viewed in any arbitrary orientations. In some cases, FIB-SEM stacks were denoised with the DenoisEM plugin[91] in FIJI[92].

## Segmentation and analysis
Membranes were manually segmented in Microscopy Image Browser (MIB)[93]. Rough segmentations of the axonemes were also performed in MIB by local thresholding. Segmentation of microtubules was performed by skeleton tracing in Knossos[94]. All segmentation data were centrally loaded into the CellSketch viewer within the Album software[95] according to our recently published protocol[96]. This workflow was also used to generate data on microtubule lengths which were plotted with PlotsOfData[97]. The plots showing the displacement of microtubules from the cilia membrane were generated by calculating the distances of points along each microtubule to the membrane using the Euclidean distance transform. These distances were then smoothed using a Gaussian filter with sigma = 3 and plotted against the point index along the microtubule. Segmentation masks were rendered with Blender (www.blender.org) according to our protocol or ORS Dragonfly (www.theobjects.com/dragonfly/index.html). For the rendering in Fig. 1c and Supplementary Movie 1, we segmented the cells organelles with Autocontext in ilastik[98].

## SEM imaging
Isolated mouse islets were fixed with 4% formaldehyde in 100 mM phosphate buffer, followed by post-fixation in modified Karnovsky fixative (2% glutaraldehyde and 2% formaldehyde in 100 mM phosphate buffer). The samples were washed $2 \times 5$ min with PBS and $3 \times 5$ min with bi-distilled water and post-fixed in 1% osmiumtetroxide in water for 2 h on ice, followed by washes in water ($6 \times 5$ min), dehydration in a graded series of ethanol/water mixtures up to pure ethanol (30%, 50%, 70%, 96% 15 min each, and $3 \times 100$% on molecular sieve, 30 min each) and critical point drying using a Leica CPD300. Dried samples (complete islets or broken specimens) were mounted on 12 mm aluminum stubs using conductive carbon tabs. To increase contrast and conductivity, samples were sputter coated with gold (BAL-TEC SCD 050 sputter coater, settings: 60 s, with 60 mA, at 5 cm working distance). Finally, samples were imaged with a JSM 7500 F scanning electron microscope (JEOL, Freising, Germany) running at 5 kV (using the lower SE-detector and working distances between 3 and 8 mm).

## Confocal microscopy and SIM of mouse pancreas sections
Pancreata from two C57BL/6 mice were dissected and immersion fixed in 4% PFA. After cryoprotection in sucrose at 4 °C overnight the specimens were embedded in TissueTek and plunge-frozen in liquid nitrogen. Cryosections were cut on a NX-70 cryostat (Thermo Fisher). Immunofluorescence staining was done on 10 or 40 μm thick cryosections of mouse pancreatic tissue. The following primary antibodies were used: insulin conjugated with Alexa 488 (Thermo Fisher, Cat. No. 53-9769-82), EPCAM (Abcam, Cat. No. ab282457), beta III tubulin (biotechne, Cat. No. MAB1195), ARL13B (Proteintech, Cat. No. 17711-1-AP), synapsin 1 (Synaptic Systems, Cat. No. 106 009), vAChT (Synaptic Systems, Cat. No. 139 105), synaptophysin 1 (Synaptic Systems, Cat. No. 101 011). Secondary antibodies were goat-raised and conjugated with Alexa Fluor dyes. Confocal images were acquired using a Nikon C2+ confocal microscope with a 60x oil immersion objective. SIM images were acquired with a Nikon SIM-E microscope with a 100× oil immersion objective. Images were obtained from pancreas samples from two individual mice. Axo-ciliary synapses in samples from two individual mice were manually quantified in 3D confocal stacks with FIJI. Overall, 1700 beta cell primary cilia in 24 islets were assessed.

## U-ExM of mouse pancreas
Whole mouse pancreas from adult C57BL/6 mice were obtained from Charles River Italy and fixed overnight in 4% PFA at 4 °C. U-ExM was performed on mouse pancreas by two different methods described below, those being whole tissue expansion and expansion of sectioned tissue.

*Whole tissue expansion, adapted from ref.* 99. After overnight 4% PFA fixation, whole pancreata were incubated in 2% acrylamide, 1.4% formaldehyde for 72 h in 1.5 ml tubes. After the crosslinking prevention, the acrylamide/formaldehyde solution was removed, and 500 μl of inactivated monomer solution (containing 0.1% triton) was added to submerge the pancreas, and incubated overnight at 4 °C. The following day, the inactivated monomer solution was removed. 500 μl of monomer solution supplemented with APS and TEMED (0.05% final concentration), as well as 0.1% triton was added and the samples were incubated for 1 h at 4 °C, followed by 2 h at 37 °C in a 1.5 ml tube.

Following gelation, the samples were removed from the 1.5 ml tube with a fishing hook, and placed in a 6-well plate containing 1 ml of denaturation buffer. Samples were shaken at room temperature for 10 min, the 6-well plate was wrapped with parafilm, and incubated for 72 h at 70 °C. Following denaturation, the gels were added to a 145 mm dish containing ddH$_2$O for 30 min to initiate expansion. The gels were washed once for 30 min with ddH$_2$O, and incubated overnight in ddH$_2$O to reach full expansion.

The following day, the gels were shrunk by washing the sample twice in 1× PBS for 15 min. Upon shrinking, the gels were incubated with the desired primary antibodies in PBS-BSA 2%, for 72 h at 37 °C while shaking. After primary antibody labeling, the gels were washed three times for 15 min and were then incubated with secondary antibodies for 48 h at 37 °C while shaking. The primary antibodies used for U-ExM were mouse-anti-insulin (Sigma, Cat. No. I2018), mouse-anti-acetylated tubulin (Sigma, Cat. No. T7451), rabbit-anti-Arl13b (Proteintech, Cat. No. 17711-1-AP), rabbit-anti-DNAI1 (Proteintech Cat. No. 12756-1-AP), rabbit-anti-CCDC39 (Sigma, Cat. No. HPA035364), rabbit-anti-GAS8 (Sigma, Cat. No. HPA041311), rabbit-anti-KIF9 (Sigma, Cat. No. HPA022033).

For NHS Ester stained gels, a stock concentration of NHS Ester 405 (Thermo Fisher, Cat. No. A30000) of 1 mg/ml was diluted to a 20 μg/mL working solution in 1× PBS. Samples were incubated for 3 h at room temperature while gently shaking. Samples were then washed in 1× PBS, 5 times by 5 min.

After secondary antibody washing and/or NHS staining, gels were washed twice for 30 min with ddH$_2$O and then incubated in water until imaging.

*Expansion of sectioned tissue adapted from ref.* 54. After overnight 4% PFA fixation, samples were washed four times, for 5 min, in 1× PBS. Whole pancreata were then incubated in 30% sucrose solution for 12 h at 4 °C. Immediately afterward, the pancreata were incubated in 15% sucrose for 12 h at 4 °C. The following morning, the pancreata were removed from 15% sucrose, and the excess solution was blotted away with filter paper. A cryomold was subsequently filled with a thin layer

of OCT, the pancreata were placed inside of a cryomold. The cryomold was placed on dry ice and was filled with OCT. Upon freezing of the OCT, the cryomold was collected and immediately transferred to a Leica CM1950 cryostat for tissue sectioning. Mouse pancreata were transversely sectioned in 20 μm increments, and individually collected on poly-d-lysine (PDL) coated, 12 mm coverslips. Sections were stored in a sealed box at −80 °C overnight, and expansion proceeded the next day as described in section 8.

### Fluorescence microscopy of U-ExM gels

Expanded gel quarters were placed in a dish and cut into an approximately 1.5 cm × 1.5 cm square piece. The gel piece was then placed on a 24 mm coverslip and added to a 35 mm imaging chamber. The chamber was placed on a Zeiss LSM980-NLO Airyscan 2 confocal microscope, and the side containing cells or tissue was identified in widefield mode. Upon correct siding, the gel was removed from the chamber, and gently blotted to remove excess water, and placed tissue side facing down on PDL coated coverslip and added back to the imaging chamber. Z-stacks were acquired in Airyscan mode with maximum resolution selected, a 1.4NA, 63×, oil immersion objective, and a z-step size of 0.15 μm.

### U-ExM image quantification

Z-stacks were loaded in Fiji. For length measurements, the simple neurite tracer plugin[100] was utilized to trace the tubulin fluorescence signal of the cilium from the bottom of the basal body in a semi-automated manner, as previously described for IFT trains[53]. Cilia length measurements were plotted and statistically analyzed in Prism 9.

### Quantification of U-ExM isotropy

Two methods of validation were used to assess expansion isotropy. (1) Measuring the nuclear cross-sectional (NCS) area of unexpanded vs. expanded beta cells and (2) measuring the length and width dimensions of daughter centrioles in beta cells by FIB-SEM and U-ExM. In the case of measuring the NCS in non-expanded samples, mouse pancreas sections were stained with NucBlu (Thermo Fisher, Cat. No. R37605), and the area of 25 nuclei was manually measured in Fiji. One slice was used for each measurement. For expanded samples, 25 beta-cell nuclei were measured in previously imaged gels, were identified by NHS ester staining, and the area was manually measured in Fiji. One slice was used for each measurement. NCS was calculated as previously described in ref. 101, by taking the square root of the measured nucleus area and dividing by the measured expansion factor of the gel (e.g. 4.2 times expansion).

To measure centriole dimensions, FIB-SEM micrographs were opened in IMOD, and daughter centrioles were identified by the absence of a cilium, and the length and width were manually measured. In expanded samples, daughter centrioles in beta cells were identified by the absence of a cilium by acetylated tubulin staining, and the length and width were manually measured in Fiji by taking the full width at half maximum values of fluorescence signal plot profiles. Both NCS and daughter centriole values were plotted and statistically analyzed in Prism 9.

Given the homogenous width of the proximal basal body, these measurements were used as a biological ruler to normalize for the expansion factor and scale the images. The true measured width was divided by 225 nm to obtain the expansion factor (e.g. a true measured width of 968 nm/225 nm = and expansion factor of approximately 4.3).

## Data availability

The FIB-SEM data used in this study for segmentation of the mouse beta cell cilia axoneme are deposited here: https://openorganelle. janelia.org/datasets/jrc_mus-pancreas-3 The open-access raw FIB-SEM datasets used for the investigation of cilia restriction and interaction can be found at: https://openorganelle.janelia.org/datasets/jrc_mus-pancreas-1 and https://openorganelle.janelia.org/datasets/jrc_mus-pancreas-2. Source data are provided with this paper.

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

## Acknowledgements

We thank the electron microscopy facility of MPI-CBG for their services, as well as the National Facility for Light Imaging and Tissue Processing Unit at Human Technopole. We thank Katja Pfriem for administrative assistance. We thank members of the PLID, especially Lisa-Marie Kretschmar, for valuable feedback. We thank Olof Idevall-Hagren (Uppsala University, Sweden) for critical comments on the manuscript. We thank Isabel Espinosa-Medina, Wei-Ping Li, Zhiyuan Lu, Wei Qiu, Eric Trautman, Stephan Preibisch, Davis Bennett, Yurii Zubov, Rebecca Vorimo, Aubrey Weigel, and the CellMap Project Team (all HHMI/Janelia) for generating and sharing the FIB-SEM dataset of P7 mouse pancreas. This work was supported with funds to M Solimena from the German Center for Diabetes Research (DZD e.V.) by the German Ministry for Education and Research (BMBF), from the German-Israeli Foundation for Scientific Research and Development (GIF) (grant I-1429-201.2/2017), from the German Research Foundation (DFG) jointly with the Agence Nationale de la Recherche (ANR) (grant SO 818/6-1), and from the Innovative Medicines Initiative 2 Joint Undertaking under grant agreements no. 115881 (RHAPSODY) and no. 115797 (INNODIA), which include financial contributions from the European Union's Framework Programme Horizon 2020, EFPIA, and the Swiss State Secretariat for Education, Research and Innovation (SERI) under contract number 16.0097, as well as JDRF International and The Leona M. and Harry B. Helmsley Charitable Trust. This project received funding from the European Union Funded Network "INTERCEPT-T2D" 101095433, which includes financial contributions from the European Union's Framework Program Horizon 2020. Views and opinions expressed are however those of the authors only and don't necessarily reflect those of the funding agencies. Neither the European Union nor any of the granting authorities can be held responsible for them. G Pigino is supported by Human Technopole and the European Research Council (ERC) under the European Union's Horizon 2020 research and innovation program (grant agreement no. 819826). AM was the recipient of a MeDDrive grant from the Carl Gustav Carus Faculty of Medicine at TU Dresden, a grant from the Deutsche Diabetes Gesellschaft (DDG), and a DZD grant (82DZD08E1G). NK was the recipient of

EMBO Fellowship ALTF 537-2021. CSX, SP, and HFH are supported by the Howard Hughes Medical Institute. M Seliskar was supported by the International Federation of Medical Students Association (IFMSA). D Sulaymankhil was supported by the Research Experience Program of TU Dresden. SAD was supported by a DIGS-BB summer research internship. D Schmidt was funded by HELMHOLTZ IMAGING, a platform of the Helmholtz Information & Data Science Incubator.

## Author contributions

Project design and supervision: A.M., N.K., G.P., M. Solimena; Data acquisition and analysis: A.M., N.K., S.P., L.E.G.G., D.S., O.T., S.A.D., M. Seliskar, T.K., D.S., C.S.X., H.F.H.; Patient recruitment and surgery: J.W. and M.D.; Sample collection and processing: E.S., N.K., D.F., S.K., C.M., H.M.; Writing-Original Draft: A.M., N.K., G.P., M. Solimena; Writing-Review and Editing: All coauthors.

## Funding

## Competing interests

C.S.X. and H.F.H. are the inventors of a US patent assigned to HHMI for the enhanced FIB-SEM systems used in this work: Xu, C.S., Hayworth, K.J., and Hess, H.F. (2020) Enhanced FIB-SEM systems for large-volume 3D imaging. US Patent 10,600,615, 24 Mar. 2020. The other authors declare no competing interests.
