## [Peer Review File · Nature Communications]

REVIEWER COMMENTS

Reviewer #1 (Remarks to the Author):

In this manuscript the authors analyzed 3D structure of pancreatic islet primary cilia from the beta cell of mouse and human. Although the beta cell is key for pathogenesis of diabetes and their cilia are known to be the cellular antenna, the role of cilia and its mechanism is still to be investigate. For example, while it is called primary cilia, motility was reported recently (Cho et al. 2022 Sci. Adv.). Precise description of 3D arrangement and morphology is obviously important for understanding of these islet cilia.

The authors provided wide overview of primary cilia from pancreatic islet tissue for the first time, employing FIB-SEM, serial section electron tomography and expansion microscopy. Their novel information includes multiple ciliation in the individual cell, variation of ciliary pockets around the basal body and the axoneme, variety of 9+0 arrangement of doublet and singlet microtubules, and contact between cilia and between the cilium and another cell (which they call synapse). All these new findings will inspire wide variety of cell biologists, especially those who are interested in diabetes. This reviewer recommends publication of this manuscript after the following points are addressed.

Fig.2a and p.4, left line10: Fig.2a is not presenting enough triplet-doublet transition. It should be indicated where the C-tubule is terminated. Section from a tomogram to show triplets will help.

Fig.2e right and p.4, right line 8: the average of A-tubule and B-tubule length in Fig.2e seem almost the same, while in the text the authors stated A and B-tubules have in average 2.05 and 1.20 microns. One might be wrong.

Fig.6a and p.7, right line10: Since the outline of the cell is not shown in Fig.6a, this reviewer cannot judge whether the two cilia are from the same cell or two adjacent cells (in the left panel, it is obviously from one cell, though).

Fig.7a and p.11, left line2: It is not clear where the synaptic vesicles in fig.7a. Could you indicate them? It is also important to know how the authors define synapse from the tomograms. Are there any criteria, other than short distance between cilia and cell membrane, and vesicles around that area?

Reviewer #2 (Remarks to the Author):

In this manuscript, Andreas Muller and colleagues utilize advanced imaging technologies to comprehensively characterize the tissue distribution and interactions of primary cilia in the pancreas, particularly focusing on their ultrastructure in beta cells, which play a pivotal role in regulating glucose levels through insulin secretion.

The authors initially employ Volume Electron Microscopy imaging technologies, including FIB-SEM and ssET, along with 3D image segmentation on both mouse and human samples. This allows them to examine and compare the 3D organization of axonemal microtubules along the length of the primary cilium. They demonstrate that primary cilia within pancreatic tissue structures exhibit a similar architecture to those in other tissues, featuring a 9+0 architecture closer to the basal body, transitioning progressively towards the cilia tip with an arrangement of axonemal microtubules in singlets moving toward the center of the axoneme. Additionally, the data indicate that primary cilia in the pancreas of both mice and humans extend doublet microtubules differently, leaving a significant portion of the axonemal length composed solely of singlet microtubules (α -tubule).

Furthermore, the authors analyze primary cilia within islets and observe that they are either situated in deep ciliary pockets or found between cells, sometimes in contact with other cilia or in proximity to cell membranes. In certain instances, cilia are enclosed in membrane-delimited tunnels within adjacent cells. These observations suggest the potential for primary cilia to sense the local environment and communicate with surrounding cells within pancreatic tissue. Lastly, the authors illustrate that primary cilia can contact neuronal projections within pancreatic tissue, resembling recently described primary cilia-neuron synapses from the Clapham lab (Shu-Hsien et al. *Cell* 2022).

Overall, this manuscript presents a series of descriptive yet fundamental observations regarding the architecture of primary cilia in pancreatic cells, warranting publication in *Nature Communications* following revision. Previous studies have indicated that beta cell primary cilia exhibit oscillatory movement and may contain protein complexes involved in cilia motility (Jung Hoon et al. *Science Advances* 2022), including components of the central pair, which are not evident in the 3D reconstructions presented in this paper. Therefore, the data provided represent an important addition to the existing literature, offering novel insights and enhanced clarity on the potential motility behavior of primary cilia in pancreatic cells.

A significant strength of the manuscript lies in the utilization of complex technical approaches, enabling clear conclusions regarding the 3D architecture of primary cilia in the pancreas. The detailed analysis of FIB-SEM datasets, collected at 4 nm resolution through the Cell Map project at HHMI Janelia, as well as the acquisition and analysis of consecutive ssET tomographic volumes, are time-consuming tasks requiring a high level of expertise and knowledge of tissue electron

microscopy features. It would be pertinent for the authors to provide evidence from ssET tomograms regarding the presence of dynein arms complexes, which are characteristic of motile cilia, and to comment on their lack of if they are not visible.

Conversely, the application of U-ExM microscopy methodology to analyze % ciliation and length appears largely unnecessary and raises more questions than answers. Similar observations of cilia length and distribution could have been achieved using immunohistochemistry approaches on tissue sections followed by confocal or other super-resolution microscopy techniques such as Airyscan or SIM imaging. The manuscript lacks information regarding whether there was isotropic expansion in pancreatic tissue and details of the factors used to convert measured lengths in the expanded tissue to reported absolute lengths and their derivation. To bolster the conclusions, a comparison of primary cilia length measurements in tissue sections by IHC to those obtained by U-ExM is recommended. Analyzing multiple human tissue sections and large areas via IHC could facilitate the examination of cilia length in patient samples of varying ages and sexes, potentially explaining the observed variability in human samples (Fig. 2e) and further strengthening the manuscript. A valuable application of U-ExM would be to determine if DNAH5 and other motility complex proteins, as shown to be present in the study by Jung Hoon et al. (Science Advances 2022), label primary cilia in islets and exhibit a 96 nm repeat localization as seen in other motile cilia.

Reviewer #3 (Remarks to the Author):

Comments

The authors use three different state of the art 3D ultrastructural imaging modalities to investigate the organization of the axoneme within the primary cilia of beta cells within its tissue context. Due to the suggested importance of primary cilia in the development of diabetes, a complete understanding of these structures is critical to developing approaches to modulate their function in pathological conditions.

There are four primary conclusions drawn: 1) there is disorganization of the axoneme in beta cell primary cilia, 2) primary cilia are confined either by pockets or surrounding cells, 3) primary cilia from alpha or beta cells interact with neighboring cells and/or their cilia, and 4) beta cell primary cilia can form synapses with pancreatic nerves. While disorganized axonemes in human beta cell primary cilia have been shown previously (Polino et al, PNAS 2023), the other three conclusions add significantly to the structural basis for primary cilia signaling between beta cells and adjacent cell types. The images and videos are beautiful and the manuscript is largely well written. I only have several relatively minor concerns that should be addressed.

The authors suggest that primary cilia can push into adjacent cells and bend their plasma membranes. However, it is also possible that plasma membranes may pull the cilia into the adjacent cells. These two possibilities cannot be distinguished from single time point EM images.

Introduction, first paragraph: it is stated that cilia are proposed to be involved in the pathogenesis of diabetes. Please provide a reference to support this.

Introduction, second paragraph: it is first stated that the axoneme is formed by nine microtubule triplets. However, soon after, it states that axonemes generally have either nine microtubule doublets with two central microtubules (9+2, motile cilia) or nine microtubule doublets with no central microtubules (9+0, primary cilia). These two statements seem contradictory. Please clarify.

Introduction, third paragraph: it is suggested that SEM does not offer 3D information which is somewhat misleading. Please clarify.

Introduction, third paragraph: The authors state that primary beta cell cilia ultrastructure has only been assessed by 2D methods. However, the lead author recently published another paper in J Cell Biology (2021) which contains a full 3D renderings of beta cell primary cilia obtained from volume electron microscopy images. Additionally, the authors have recently published volume microscopy images of primary beta cell cilia in Nature (2021), and another publication in 2017 (Gan et al. J Cell Science) also contains volume electron microscopy images of beta cell primary cilia. This statement should be amended and the relevant references included.

Figure 2C-D: The methods for how these data were obtained should be included. Were distance transforms between the segmented microtubules and membranes used here?

Reviewer #4 (Remarks to the Author):

The authors perform elegant electron and expansion microscopy work to describe the ultra-structural arrangement of primary cilia in mouse and human islets.

They find, and as other studies have suggested (i.e., Ref 35), that the organization of the primary cilia follows a disorganized 9+0 arrangement. Besides establishing what are likely the first high-resolution measurements of cilia morphology in islet alpha/beta cells, the most striking finding is the observation that cilia form "synapses" with islet autonomic neurons. These cilia-nerve contact regions have the appearance of close/dense membrane-associated structures and potential secretory granules.

I think that this study provides important and very relevant structural information to the field; however, the study falls short in demonstrating how these aspects correlate to islet cell function and/or disease states. I can fully appreciate the complexity and time/resources-consuming nature of the techniques applied in this manuscript, but I feel that more experimentation is needed to significantly show the impact of the discoveries here. For example, what would the microtubule arrangement look like in T2D or T1D cells? I understand that human tissue availability might be a complicating factor here, but at least experiments with a mouse model where glucose signaling is impaired/T2D (i.e., db/db, high-fat diet, acute S961 exposure, NOD mouse). This would also allow for in vitro experimentation using ExM (more on this below) and freeze-fraction of isolated islets from these models followed by high-res EM.

I also think that more data is needed to support the statement of a cilia-neuron synapse existing within the islet microenvironment. Given the potential impact of this discovery, I think the authors must show (using ExM, for example) that classical pre- and/or post-synaptic markers are present in these regions. Are basoon and psd95 expressed here? What type of nerves form these connections (sympathetic or parasympathetic?) and do you find GAD65/67, VACHT, Substance P, or norepinephrine associated with these regions? The authors mention that the nerves "contacted the plasma membranes of islets cells" but don't mention whether these contact regions have "synapse-like" characteristics as well. This is an important question in the field and that this study is uniquely positioned to answer. Comparing cell vs cilia neuron-contact regions would be important to classify differences in neuronal contacts and perhaps demonstrate the uniqueness of the cilia-neuron region.

Finally, I think that the text needs more guardrails to accommodate alternative explanations for their observations. For example, in the discussion referring to the microtubule organization, the authors write "While these data do not fully rule out the possibility that previously observed phenomena (Ref 35) cannot be mediated by..." and then explain that cilia movement is regulated by brownian motion. However, the studies described in Ref 35 clearly show that glucose regulates ciliary motion and implicate dyneins/atp/glucose in this behavior. The experiments suggested above would address this point and contextualize this study with the previous findings and potentially expand our knowledge in this exciting area of islet biology.

We thank the reviewers for their general enthusiasm and insightful comments. We have resolved most of their questions experimentally. Mainly, we have focused on the two major and most pressing questions: First, the presence of motility components in beta cell primary cilia and second, the characterization of axo-ciliary synapses in islets. For both topics we have made major advances, which we have added to the manuscript and are explained below.

REVIEWER COMMENTS

Reviewer #1 (Remarks to the Author):

In this manuscript the authors analyzed 3D structure of pancreatic islet primary cilia from the beta cell of mouse and human. Although the beta cell is key for pathogenesis of diabetes and their cilia are known to be the cellular antenna, the role of cilia and its mechanism is still to be investigated. For example, while it is called primary cilia, motility was reported recently (Cho et al. 2022 Sci. Adv.). Precise description of 3D arrangement and morphology is obviously important for understanding of these islet cilia.

The authors provided wide overview of primary cilia from pancreatic islet tissue for the first time, employing FIB-SEM, serial section electron tomography and expansion microscopy. Their novel information includes multiple ciliation in the individual cell, variation of ciliary pockets around the basal body and the axoneme, variety of 9+0 arrangement of doublet and singlet microtubules, and contact between cilia and between the cilium and another cell (which they call synapse). All these new findings will inspire wide variety of cell biologists, especially those who are interested in diabetes. This reviewer recommends publication of this manuscript after the following points are addressed.

Fig.2a and p.4, left line10: Fig.2a is not presenting enough triplet-doublet transition. It should be indicated where the C-tubule is terminated. Section from a tomogram to show triplets will help.

Thank you for this comment. We added a FIB slice with the side view of the mouse cilium, in which the termination of the basal body, the axoneme and a distal appendage are well visible to Figure 2.

Fig.2e right and p.4, right line 8: the average of A-tubule and B-tubule length in Fig.2e seem almost the same, while in the text the authors stated A and B-tubules have in average 2.05 and 1.20 microns. One might be wrong.

Thanks for pointing this mistake to our attention. The numbers in the text are correct and we have corrected the graph accordingly.

Fig.6a and p.7, right line10: Since the outline of the cell is not shown in Fig.6a, this reviewer cannot judge whether the two cilia are from the same cell or two adjacent cells (in the left panel, it is obviously from one cell, though).

The reviewer is correct that the method for detecting multiple cilia per cell in UExM images is missing. Thus we have added Supplementary Figure 5 to display how we use NHS Ester staining to demarcate the volume of a single cell in 3D, or measure the distance between basal bodies.

Fig.7a and p.11, left line2: It is not clear where the synaptic vesicles in fig.7a. Could you indicate them? It is also important to know how the authors define synapse from the tomograms. Are there any criteria, other than short distance between cilia and cell membrane, and vesicles around that area?

This is a great question. We have changed Figure 7 (which is now new Figure 8) and highlighted the vesicles in a new panel and put next to it a magnification of the raw image. We did the same also for the likely vesicle fusing with the axonal membrane. Structurally, we define the presynapse similar to Sheu et al., by the presence of small electron lucent vesicles close to the contact and instances of vesicle fusion as seen in Fig 8a". To investigate this further (also in response to reviewer 4) we performed immunolabelings against presynaptic markers and found that on the neuronal side synapsin 1 is enriched at contact sites with cilia. We furthermore have identified acetylcholine to be present at these sites by stainings for vAChT. These are now the new Figures 9 and Supp. Figures 11 and 12.

Reviewer #2 (Remarks to the Author):

In this manuscript, Andreas Muller and colleagues utilize advanced imaging technologies to comprehensively characterize the tissue distribution and interactions of primary cilia in the pancreas, particularly focusing on their ultrastructure in beta cells, which play a pivotal role in regulating glucose levels through insulin secretion.

The authors initially employ Volume Electron Microscopy imaging technologies, including FIB-SEM and ssET, along with 3D image segmentation on both mouse and human samples. This allows them to examine and compare the 3D organization of axonemal microtubules along the length of the primary cilium. They demonstrate that primary cilia within pancreatic tissue structures exhibit a similar architecture to those in other tissues, featuring a 9+0 architecture closer to the basal body, transitioning progressively towards the cilia tip with an arrangement of axonemal microtubules in singlets moving toward the center of the axoneme. Additionally, the data indicate that primary cilia in the pancreas of both mice and humans extend doublet microtubules differently, leaving a significant portion of the axonemal length composed solely of singlet microtubules (α -tubule).

Furthermore, the authors analyze primary cilia within islets and observe that they are either situated in deep ciliary pockets or found between cells, sometimes in contact with other cilia or in proximity to cell membranes. In certain instances, cilia are enclosed in membrane-delimited tunnels within adjacent cells. These observations suggest the potential for primary cilia to sense the local environment and communicate with surrounding cells within pancreatic tissue. Lastly, the authors illustrate that primary cilia can contact neuronal projections within pancreatic tissue, resembling recently described primary cilia-neuron synapses from the Clapham lab (Shu-Hsien et al. Cell 2022).

Overall, this manuscript presents a series of descriptive yet fundamental observations regarding the architecture of primary cilia in pancreatic cells, warranting publication in Nature Communications following revision. Previous studies have indicated that beta cell primary cilia exhibit oscillatory movement and may contain protein complexes involved in cilia motility

(Jung Hoon et al. Science Advances 2022), including components of the central pair, which are not evident in the 3D reconstructions presented in this paper. Therefore, the data provided represent an important addition to the existing literature, offering novel insights and enhanced clarity on the potential motility behavior of primary cilia in pancreatic cells.

A significant strength of the manuscript lies in the utilization of complex technical approaches, enabling clear conclusions regarding the 3D architecture of primary cilia in the pancreas. The detailed analysis of FIB-SEM datasets, collected at 4 nm resolution through the Cell Map project at HHMI Janelia, as well as the acquisition and analysis of consecutive ssET tomographic volumes, are time-consuming tasks requiring a high level of expertise and knowledge of tissue electron microscopy features. It would be pertinent for the authors to provide evidence from ssET tomograms regarding the presence of dynein arms complexes, which are characteristic of motile cilia, and to comment on their lack of if they are not visible. Thank you for the detailed positive assessment of our work.

The last point is very interesting. When we checked for the presence of electron dense structures decorating the microtubules in ssET data we could not find much and especially not repetitive structures. A cross section of the human cilium can be found in the new Figure 3b. We used expansion microscopy to investigate the presence of dynein arms and other motility components also in response to the last question of the reviewer. These findings are now summarized in Figure 3 and the corresponding text.

Conversely, the application of U-ExM microscopy methodology to analyze % ciliation and length appears largely unnecessary and raises more questions than answers. Similar observations of cilia length and distribution could have been achieved using immunohistochemistry approaches on tissue sections followed by confocal or other super-resolution microscopy techniques such as Airyscan or SIM imaging.

There several reasons why we believe that the use of U-ExM is not only not unnecessary, but actually required to properly quantify ciliated cells and their interactions with other cells:

- Post-expansion staining usually improves the accessibility to potential epitopes for antibodies,
- It increases the visibility of details, such as the localization of the basal bodies with respect to the cell membrane, which allows us to show the presence of multiciliated beta cells. Or the contacts between two adjacent cilia, which could have been mistaken for a single cilium in non expanded samples imaged by confocal microscopy or SIM imaging.

As a consequence, U-ExM was key to perform more reliable quantifications of even simple parameters, such as ciliary number and length, in addition to more advanced features, such as cilia to cilia or cilia to cell interactions and localization of specific protein complexes in the ciliary structure.

The manuscript lacks information regarding whether there was isotropic expansion in pancreatic tissue and details of the factors used to convert measured lengths in the expanded tissue to reported absolute lengths and their derivation.

To address this valid concern of the reviewer, we validated the isotropy in the U-ExM of pancreatic sections by analysis nucleus cross-sectional (NCS) area in beta cells pre and post expansion (the data is shown in a new Supplementary Figure 3 a,b,c), as well as the length and width dimensions of the daughter centriole by EM and U-ExM (new Supplemental Figure 3 d,e,f). We found no statistical difference in the expansion factor corrected NCS area, or daughter centriole dimensions, indicating general isotropy in the expansion of pancreas sections. We provide a detailed explanation of the analysis in a new chapter of the Materials and Methods section.

We apologize for not being more clear in regards to reporting the expansion factor in our images. The expansion factor was normalized to the length of the proximal basal body, which as shown by the new experiment measuring expansion isotropy, is a reliable biological ruler. All scale bars are reported in the biological size, taken by measuring the true length and dividing by the measured 225 nm of the proximal basal body. A more detailed explanation has been added to the Materials and Methods.

To bolster the conclusions, a comparison of primary cilia length measurements in tissue sections by IHC to those obtained by U-ExM is recommended.

The reviewer raises a good point. Our U-ExM collected beta cell ciliary length distribution fits previously reported 3- 10 μm beta cell ciliary length values collected by immunofluorescence (GM Sanchez et. al. 2022, JCB). To further address this point, we have performed an analysis on expansion isotropy using the daughter centriole measured by EM as a ruler, and found no statistical difference between U-ExM and EM, when corrected for the expansion factor of the gel (Supplementary Figure 3). A more detailed explanation has been added to the Materials and Methods.

Analyzing multiple human tissue sections and large areas via IHC could facilitate the examination of cilia length in patient samples of varying ages and sexes, potentially explaining the observed variability in human samples (Fig. 2e) and further strengthening the manuscript.

We agree with the reviewer that the analysis of cilia in samples from patients, and particularly the potential exploration of samples representing varying ages and sexes could be very valuable. Such investigations could represent a unique opportunity for advancing primary cilia research and contributing insights to the field of diabetes research. However, we must emphasize the performance of these additional analyses would necessitate a very substantial investment of resources and time, extending beyond what is realistically feasible within the scope of this revised manuscript. Furthermore, given the depth and complexity of potential findings and their interpretation from a clinical perspective, we believe that such studies warrant dedicated attention in one or more independent publications, which we intend to pursue in the future .

We will certainly take into great account this reviewer idea and suggestion for further studies.

A valuable application of U-ExM would be to determine if DNAH5 and other motility complex proteins, as shown to be present in the study by Jung Hoon et al. (Science Advances 2022), label primary cilia in islets and exhibit a 96 nm repeat localization as seen in other motile cilia.

We thank the reviewer for this comment. We were intrigued by the unexpected results of the study by Jung Hoon Cho et al. (Science Advances 2022) and we already started performing some IF experiments to verify the presence of motility components in islet cells. Because our initial results were in conflict with what was observed by Cho et al., we refrained from including this data in our initial submission. We have now repeated the same stainings several times and can confidently state that DNAH5 and other motility components are indeed not present in beta cell primary cilia. Thus, we have now included this data in a main figure (new Figure 3), as well as in the text of the results and of the discussion.

Reviewer #3 (Remarks to the Author):

Comments

The authors use three different state of the art 3D ultrastructural imaging modalities to investigate the organization of the axoneme within the primary cilia of beta cells within its tissue context. Due to the suggested importance of primary cilia in the development of diabetes, a complete understanding of these structures is critical to developing approaches to modulate their function in pathological conditions.

There are four primary conclusions drawn: 1) there is disorganization of the axoneme in beta cell primary cilia, 2) primary cilia are confined either by pockets or surrounding cells, 3) primary cilia from alpha or beta cells interact with neighboring cells and/or their cilia, and 4) beta cell primary cilia can form synapses with pancreatic nerves. While disorganized axonemes in human beta cell primary cilia have been shown previously (Polino et al, PNAS 2023), the other three conclusions add significantly to the structural basis for primary cilia signaling between beta cells and adjacent cell types. The images and videos are beautiful and the manuscript is largely well written. I only have several relatively minor concerns that should be addressed.

The authors suggest that primary cilia can push into adjacent cells and bend their plasma membranes. However, it is also possible that plasma membranes may pull the cilia into the adjacent cells. These two possibilities cannot be distinguished from single time point EM images.

This is a good point. We rephrased these sentences to make it more neutral.

Introduction, first paragraph: it is stated that cilia are proposed to be involved in the pathogenesis of diabetes. Please provide a reference to support this.

We added references to this point. Notably, most data so far comes from mouse models. However, Kluth et al. show human data transcriptomic data.

Introduction, second paragraph: it is first stated that the axoneme is formed by nine microtubule triplets. However, soon after, it states that axonemes generally have either nine microtubule doublets with two central microtubules (9+2, motile cilia) or nine microtubule doublets with no central microtubules (9+0, primary cilia). These two statements seem contradictory. Please clarify.

Thank you for this comment. We have now adapted the text to make it clearer that the C-tubule terminates early in the transition zone.

Introduction, third paragraph: it is suggested that SEM does not offer 3D information which is somewhat misleading. Please clarify.

SEM does offer 3D surfaces but not the internal organization below the surfaces. This can be overcome by extracting the plasma membrane as done in Polino et al., 2023. However, this is a destructive procedure and ultimately alters the structural context. We edited this part in the introduction.

Introduction, third paragraph: The authors state that primary beta cell cilia ultrastructure has only been assessed by 2D methods. However, the lead author recently published another paper in J Cell Biology (2021) which contains a full 3D renderings of beta cell primary cilia obtained from volume electron microscopy images. Additionally, the authors have recently published volume microscopy images of primary beta cell cilia in Nature (2021), and another publication in 2017 (Gan et al. J Cell Science) also contains volume electron microscopy images of beta cell primary cilia. This statement should be amended and the relevant references included.

We added these citations. 3D information, also by us and others, was there, but the full reconstruction of the axoneme had not yet been done. This is now clearer in the text.

Figure 2C-D: The methods for how these data were obtained should be included. Were distance transforms between the segmented microtubules and membranes used here?

Yes, we added the explanation for this in the methods section.

Reviewer #4 (Remarks to the Author):

The authors perform elegant electron and expansion microscopy work to describe the ultrastructural arrangement of primary cilia in mouse and human islets.

They find, and as other studies have suggested (i.e., Ref 35), that the organization of the primary cilia follows a disorganized 9+0 arrangement. Besides establishing what are likely the first high-resolution measurements of cilia morphology in islet alpha/beta cells, the most striking finding is the observation that cilia form "synapses" with islet autonomic neurons. These cilia-nerve contact regions have the appearance of close/dense membrane-associated structures and potential secretory granules.

I think that this study provides important and very relevant structural information to the field; however, the study falls short in demonstrating how these aspects correlate to islet cell function and/or disease states. I can fully appreciate the complexity and time/resources-consuming nature of the techniques applied in this manuscript, but I feel that more experimentation is needed to significantly show the impact of the discoveries here. For example, what would the microtubule arrangement look like in T2D or T1D cells? I understand that human tissue availability might be a complicating factor here, but at least experiments with a mouse model where glucose signaling is impaired/T2D (i.e., db/db, high-fat diet, acute S961 exposure, NOD mouse). This would also allow for in vitro experimentation using ExM (more on this below) and freeze-fraction of isolated islets from these models followed by high-res EM.

This is a very interesting point since there is no structural data on human beta cell primary cilia during the progression of diabetes available. However, we feel that this question is beyond the scope of our manuscript and we have not been confident to provide the significant amount of data that would be necessary to resolve these questions in the short timeframe available for the revision of the manuscript. We therefore focused our efforts on the two other main questions raised by the reviewers: 1) the presence of motility complexes in the beta cell cilium, and 2) the characterization of beta cell axo-ciliary contacts.

I also think that more data is needed to support the statement of a cilia-neuron synapse existing within the islet microenvironment. Given the potential impact of this discovery, I think the authors must show (using ExM, for example) that classical pre- and/or post-synaptic markers are present in these regions. Are bassoon and psd95 expressed here? What type of nerves form these connections (sympathetic or parasympathetic?) and do you find GAD65/67, VACHT, Substance P, or norepinephrine associated with these regions? The authors mention that the nerves "contacted the plasma membranes of islets cells" but don't mention whether these contact regions have "synapse-like" characteristics as well. This is an important question in the field and that this study is uniquely positioned to answer. Comparing cell vs cilia neuron-contact regions would be important to classify differences in neuronal contacts and perhaps demonstrate the uniqueness of the cilia-neuron region.

We fully agree that more data is necessary to support our initial observations. We have therefore performed immunostainings for numerous presynaptic markers together with stainings for markers of primary cilia, insulin and the axonal marker beta III tubulin. We could find cases of cilia contacting nerves also in these fluorescence images by confocal and structured illumination. Furthermore, we found that in most cases synapsin 1, a marker of neuronal synaptic vesicles, was enriched close to the cilium. We tested several antibodies in order to identify the neurotransmitters involved in the axo-ciliary signaling. Stainings for SERT and DAT were not successful. However, stainings for vAChT showed signals that colocalized with synapsin 1 and were enriched at the axo-ciliary contact sites. We therefore conclude that acetylcholine is involved in this signaling pathway. Notably, all contacts observed in fluorescence contained synapsin 1 and/or vAChT, indicating the presence of synaptic vesicles. In the EM we could not see such vesicles in all cases, either because of the limited resolution/contrast or because of the young age of the mice. This sample is from a P7 pancreas, where the innervation of the pancreas might not be complete. These results of these new stainings are summarized in the new Figure 9 and Supplementary Figures 11 of the manuscript. We also tested markers for postsynaptic compartments but those stainings did not show any signal. As these markers have not been described by Sheu et al. for axo-ciliary synapses in the brain, we assume that they are not present in cilia. Instead, the cilium itself may act as a postsynaptic compartment. We added a part concerning this point to the discussion.

The question regarding the comparison between cilia-neuron and cilia-plasma membrane synapses is very interesting too. To resolve this issue it would be necessary to obtain structural as well as molecular information on synapses at the membrane and cilia. This would require correlative light and electron microscopy, which is extremely time-consuming and not feasible in the timeframe of the revision. We therefore looked at this issue by immunostainings for the membrane marker EPCAM and found that axons with vAChT

positive varicosities were in contact with beta cell membranes similar to cilia (new Supplementary Figure 12). Being the same neurotransmitter used in both synapses, the difference could be how the cilium propagates the signal and the relative spatial contiguity to the nucleus (as proposed in Sheu et al., 2022). We added some thoughts about this point in the discussion.

Finally, I think that the text needs more guardrails to accommodate alternative explanations for their observations. For example, in the discussion referring to the microtubule organization, the authors write "While these data do not fully rule out the possibility that previously observed phenomena (Ref 35) cannot be mediated by..." and then explain that cilia movement is regulated by brownian motion. However, the studies described in Ref 35 clearly show that glucose regulates ciliary motion and implicate dyneins/atp/glucose in this behavior. The experiments suggested above would address this point and contextualize this study with the previous findings and potentially expand our knowledge in this exciting area of islet biology.

We are aware that this topic is controversial. Our new data on the absence of motility components in beta cell cilia by UExM (new Figure 3), however, provides compelling evidence that beta cell cilia do not move in a motor-dependent fashion.

REVIEWER COMMENTS

Reviewer #1 (Remarks to the Author):

The authors addressed all points from this reviewer, by additional experiments, remaking presentations and editing the text. Newly made Fig.3 presents convincing comparison between Beta cells primary cilia and motile cilia, tracking component proteins of motile cilia. Now the manuscript is nearly complete for publication. The only (minor) concern of this reviewer is about Fig.2a. The authors indicate the transition zone by a white arrowhead. Due to low resolution of the figure (I believe there was not high-resolution image specially provided for the review process), it cannot show the transition from basal body triplets to axonemal doublets. As soon as seeing high resolution picture of Fig.2a and confirming the transition zone, this reviewer will strongly support publication of this manuscript.

Reviewer #2 (Remarks to the Author):

The authors have effectively addressed the majority of my suggestions and comments. Furthermore, they have provided compelling evidence supporting the existence of bona fide pancreatic cilia synapses. I recommend the article for publication without further delay.

Reviewer #3 (Remarks to the Author):

The authors have responded well to my previous minor concerns. I do not have any remaining issues with this very nice work.

Reviewer #4 (Remarks to the Author):

The authors have addressed most of the concerns raised by me and the other reviewers. The new UxM and innervation data is exciting and compelling. However, the manuscript lacks clarity regarding the sampling frequency of the cilia studied (n of structures, n islets, n of mice/human tissue) and that support the authors conclusions. In fact, Fig 9 has beautiful islet innervation and

primary cilia images; however, I can clearly see that there are several ciliary structures that are not contacting nerve structures.

While I believe that the characterization of this heterogeneity is beyond the scope of this manuscript, the authors should provide basic demographics/observation frequency of cilia-neuron contacts (see first paragraph) so that one can understand the magnitude of this data. This heterogeneity could also explain why there are apparently discrepant results between this study and ref #38. Given that a comprehensive sampling of primary cilia is not described (or presented) here -- and likely very time consuming --- one could speculate that mouse/human islets may contain both motile and non-motile cilia, and that both studies are correct. This should be addressed in the discussion and considered as a basis for future additional studies in this exciting area.

We thank the reviewers for their very positive comments on the revised manuscript and have resolved their remaining questions.

Reviewer #1 (Remarks to the Author):

The authors addressed all points from this reviewer, by additional experiments, remaking presentations and editing the text. Newly made Fig.3 presents convincing comparison between Beta cells primary cilia and motile cilia, tracking component proteins of motile cilia. Now the manuscript is nearly complete for publication. The only (minor) concern of this reviewer is about Fig.2a. The authors indicate the transition zone by a white arrowhead. Due to low resolution of the figure (I believe there was not high-resolution image specially provided for the review process), it cannot show the transition from basal body triplets to axonemal doublets. As soon as seeing high resolution picture of Fig.2a and confirming the transition zone, this reviewer will strongly support publication of this manuscript.

Thank you for this comment. The transition zone is indeed not easy to see in the 4 nm FIB-SEM data. We therefore added a side view of the ssET data of the human primary cilium at 1.307 nm voxels where this is better visible to figure 2 and explain this in the text.

Reviewer #2 (Remarks to the Author):

The authors have effectively addressed the majority of my suggestions and comments. Furthermore, they have provided compelling evidence supporting the existence of bona fide pancreatic cilia synapses. I recommend the article for publication without further delay.

Reviewer #3 (Remarks to the Author):

The authors have responded well to my previous minor concerns. I do not have any remaining issues with this very nice work.

Reviewer #4 (Remarks to the Author):

The authors have addressed most of the concerns raised by me and the other reviewers. The new UxM and innervation data is exciting and compelling. However, the manuscript lacks clarity regarding the sampling frequency of the cilia studied (n of structures, n islets, n of mice/human tissue) and that support the authors conclusions. In fact, Fig 9 has beautiful islet innervation and primary cilia images; however, I can clearly see that there are several ciliary structures that are not contacting nerve structures.

While I believe that the characterization of this heterogeneity is beyond the scope of this manuscript, the authors should provide basic demographics/observation frequency of cilia-neuron contacts (see first paragraph) so that one can understand the magnitude of this data. This heterogeneity could also explain why there are apparently discrepant results between this study and ref #38. Given that a comprehensive sampling of primary cilia is not described

(or presented) here -- and likely very time consuming --- one could speculate that mouse/human islets may contain both motile and non-motile cilia, and that both studies are correct. This should be addressed in the discussion and considered as a basis for future additional studies in this exciting area.

Thank you for these comments. We have quantified the abundance of beta cell axo-ciliary synapses and added these values to the results part. Overall we have analyzed 1700 beta cell primary cilia in 24 islets from 2 individual mice (stated now in the methods section).

We agree with the reviewer that there might be heterogeneity among beta cell cilia. However, the spatial confinement of the axonemes within a few nanometers, their connection to other cells, cilia and axons, as well as the absence of the central microtubule pair and densities attributable to dynein arms in all ultrastructurally resolved cilia strongly argue against their possible motile activity. Accordingly, none of the 200 beta cell cilia analyzed by UExM expressed any of the expected motility components, including GAS8, CCDC39, DNAI1, and KIF9. Therefore, beta cell cilia heterogeneity might instead be defined at the cilia membrane to enable different connections and signaling functions. This is a topic that needs to be elucidated in future studies. We added an additional paragraph on this topic to the discussion.

REVIEWERS' COMMENTS

Reviewer #4 (Remarks to the Author):

The authors have addressed all my concerns and now I think the manuscript is suitable for publication.